



# Enhanced western Mediterranean rainfall during past interglacials driven by North Atlantic pressure changes

Yama Dixit[1,2,3]*, Samuel Toucanne[1], Juan M. Lora[4], Christophe Fontanier[5,6,7], Virgil Pasquier[2], Lea Bonnin[2], Gwenael Jouet[1], Aradhna Tripati[1,2,4]

[1]IFREMER, Laboratoire Géophysique et enregistrement Sédimentaire, CS10070, 29280 Plouzané cedex, France
[2]Université de Brest - UMR 6538 CNRS/UBO, LGO, IUEM, 29280 Plouzané, France
[3]Earth Observatory of Singapore, Nanyang Technological University, Singapore
[4]Department of Atmospheric and Oceanic Sciences, Institute of the Environment and Sustainability, Department of Earth, Planetary, and Space Sciences, Center for Diverse Leadership in Science, University of California, Los Angeles, USA
[5]Université de Bordeaux, CNRS, Environnements et Paléo-environnements Océaniques et Continentaux, UMR 5805, F-33600 Pessac, France
[6]FORAM, Research Group, F-49140 Villevêque, France
[7]Université d'Angers, F49035 Angers, France

*Correspondence to*: Yama Dixit ydixit@ntu.edu.sg

**Abstract.** There is increasing concern with anthropogenic greenhouse gas emissions that ocean warming, in concert with summer and winter precipitation changes, will induce anoxia in multiple ocean basins. In particular the Mediterranean Sea is susceptible to severe hydrological changes. Mediterranean hydroclimate is controlled primarily by two phenomena – the latitudinal migration of the Inter-Tropical Convergence Zone and the North Atlantic climatic processes. While the former brings about the African summer monsoon rainfall the latter drives the wintertime storm tracks into the western Mediterranean. Although the hydrological changes in the eastern Mediterranean are quite well constrained, evidence of past changes in temperature and rainfall in the western Mediterranean across the past interglacials is relatively scarce. In this study, we use trace element and stable isotope composition of planktonic foraminifera from a sediment core off Corsica at the mouth of Golo river in the western Mediterranean to reconstruct variations in sea surface temperature (SST) and sea surface salinities (SSS) during the Holocene and warm periods of the past two interglacials. Our data suggest that the warm periods of the last interglacials were characterised by high river discharge and lower SSS in the northern Tyrrhenian Sea, suggesting increased winter rainfall. We find evidence that enhanced winter rainfall during periods



of precession minima and high seasonality across interglacials coincide with changes in the respective eccentricity maxima suggesting a causal link. Our model simulations for representative orbital configurations such as the mid-Holocene support increased south-westerly moisture transport into the western Mediterranean originating from the North Atlantic. We suggest that these hydrologic changes in

the western and the northern Mediterranean borderlands were a contributing factor to basin-wide anoxia in the past. Our findings offer new insights into the cause and impact of winter rainfall changes in the Mediterranean during past warm periods.

## 1 Introduction

The hydrology and ecosystems of the Mediterranean are highly vulnerable to the impacts of climate

change (Guiot and Cramer, 2016; IPCC AR4, 2014). Indeed, it is well-known that conditions throughout the Mediterranean Sea are highly sensitive to hydrological changes of the surrounding region as exemplified by the development of anoxia leading to sapropel deposition during periods of disruption in Mediterranean hydrological cycle in the past (Rohling et al., 2015 and references therein). Owing to its location, the Mediterranean Sea is sensitive to atmospheric circulation changes originating

both from the latitudinal migration of the Inter-Tropical Convergence Zone (ITCZ) and the Atlantic climatic processes that drives the wintertime Mediterranean storm track. Hydrologic changes in the eastern Mediterranean Sea today and in the past were derived by the variations in rainfall and river discharges into the basin related to enhanced summer monsoon activity over North Africa due to the movement of the ITCZ (Revel et al., 2010; Rohling et al., 2002; Rossignol-Strick, 1983; Rossignol-

Strick et al., 1985). In the western Mediterranean basin and northern borderlands, the modern hydrological changes are more dependent on the North Atlantic atmospheric circulation (Kandiano et al., 2014; Trigo et al., 2002). However, there still remains a lot of ambiguity in understanding of the rainfall variability in the western Mediterranean during the past interglacials. For example, Rohling et al., (2015) suggested that the past rainfall in the western Mediterranean was in fact the recycled

moisture from the eastern Mediterranean Sea itself. However, modern rainfall isotopic study from the western Mediterranean region show a unique characteristic signal that is between Atlantic and Eastern Mediterranean moistures (Celle-Jeanton et al., 2001).  In this light, although the hydrologic changes





during the past interglaicals in the eastern Mediterranean are well-constrained, climatic variations over the western basin and the northern Mediterranean borderlands are largely debated.

Lines of evidence for increased winter rainfall in the western Mediterranean comes from: pollen (Milner et al., 2012; Tzedakis, 2007) for the Holocene and the warm period of the last interglacial (Marine

Isotope Stage, MIS 5, (Railsback et al., 2015)); lacustrine records (Carrión, 2002; Fletcher and Sánchez Goñi, 2008; Magny et al., 2011, 2013; Peyron et al., 2011) for the Holocene; and speleothem studies (Drysdale et al., 2005; Regattieri et al., 2014; Zanchetta et al., 2007) for the last interglacial period. All these studies suggested enhancement of hydrological activity over the Mediterranean borderlands during the Holocene and MIS 5. A putative link between high seasonality and increased winter rainfall

in the Mediterranean has also been suggested for the last interglacial (Milner et al., 2012). While these studies represent an important contribution, it has been pointed out that they were unable to offer direct insights on the variability in winter rainfall mainly due to proxy limitations (Rohling et al., 2015) and also because of a lack of direct rainfall or sea surface salinity (SSS) and sea surface temperature (SST) estimates, acquired from the same archive spanning the past interglacials. Here we present a rainfall

(salinity) and temperature record for the western Mediterranean basin spanning the Holocene (MIS 1), the last (MIS 5) and penultimate (MIS 7) interglacials. We use trace element and stable isotopes of the planktonic foraminifera *Globigerina bulloides* from marine sediment core GDEC-4-2 off eastern Corsican margin (Fig. 1), and model simulations to determine the presence of extra-Mediterranean moisture source originating from the North Atlantic.

Marine sediment core GDEC-4-2 (042°31.23.2'N, 09°42.59.5'E, 492 m water depth) was recovered near the mouth of the Golo river in Corsica in the western Mediterranean basin (Fig. 1). Corsica is considered to be a highly sensitive paleoclimatic key area within the Mediterranean (Kuhlemann et al., 2008). The Golo is a short, mountainous river (maximum altitude of ca. 2700 m) and was the most important source of Pleistocene–Holocene terrigenous sediment to the adjacent margin (Calves et al.,

2013). The Golo river system is a typical example of a 'reactive' sediment routing system (sensu Allen, 2008), as it responds simultaneously to climate-driven changes and is therefore well-suited for studying the 'source to sink' dynamics and hydrology(Calves et al., 2013; Toucanne et al., 2015).



The hydrography at GDEC-4-2 site is mainly influenced by the Levantine Intermediate Water (LIW) circulation (from ca. 200 to 600-1000 m water depth) (Toucanne et al., 2012). The LIW is formed in the Levantine Basin (eastern Mediterranean) in a permanent large-scale cyclonic Rhodes gyre through summer evaporation and winter cooling (i.e. buoyancy loss; Lascaratos et al., 1999; Malanotte-Rizzoli

et al., 2003; Robinson et al., 1992). It forms the major water mass flowing from the east to west, along with the Aegean and Adriatic water contributions. In the northern Tyrrhenian Sea, a portion of the LIW flows northwards through the Corsica Trough, while the other part flows southwards to the Sardinia Channel, then along the western slope of Sardinia and Corsica before its intrusion into Ligurian Sea. The LIW contributes to the Western Mediterranean Deep Water production after reaching the Gulf of

Lion and both water masses contribute to ca. 80% and 20% to the Mediterranean Outflow water (MOW), respectively (Pinardi and Masetti, 2000).

## 2 Materials and Methods

### 2.1 Sediment core GDEC4-2

The sediment core GDEC-4-2 is from the upper continental slope on the eastern Corsica margin, directly

off the Golo River basin (see Toucanne et al., 2015 for details). The Golo system is a highly efficient sediment routing system characterized by a 89 km long mountainous river and a drainage basin of ca 1214 km$^2$ (Calves et al., 2013) and is ideally located on the way of North Atlantic depression into the Western Mediterranean Sea (Bengtsson et al., 2009; Hoskins and Hodges, 2002; Kutzbach et al., 2014), such that the sediment core acquired from this site faithfully captures the variation in winter storm tracks in the

sediments. The studied interval is composed of hemipelagic sediments, mainly silty-clay carbonate rich intervals.

### 2.2 Stable isotope analyses

The planktonic foraminifera *G. bulloides*, that occurs virtually continuously throughout the interglacial intervals of core GDEC-4-2, was selected for the stable oxygen and carbon isotope analysis using 15–20

specimens from the 250–300 μm size fraction. For MIS 5e (Railsback et al., 2015) we resampled *G. bulloides* and reanalysed at higher resolution and for the Holocene and MIS 7c and 7e, we used stable



isotope data from Toucanne et al. (2015). Oxygen and carbon isotope ratios were measured using Thermo Scientific Delta V plus Isotope Ratio Mass Spectrometer fitted with a GasBench II preparation and introduction device, operated by Pôle Spectrométrie Océan (PSO, IFREMER, IUEM, CNRS), located at the Institut Universitaire Européen de la Mer (IUEM / UBO) at Plouzané, France. Analytical precision of

powdered carbonate standards is 0.2‰ for $\delta^{18}O$ and 0.1‰ for $\delta^{13}C$. The stable isotopes are expressed as $\delta^{18}O$ and $\delta^{13}C$ relative to the Vienna Pee Dee Belemnite (V-PDB) standard.

## 2.2 Trace element analyses

Trace element ratios have been measured on 5 to 20 specimens of *G. bulloides* were picked from the 250-300 μm fraction for each analysis. *G. bulloides* occurred continuously throughout the sediment core as

compared to other planktonic foraminifera such as *G. ruber* and *N. pachyderma*, which appeared sporadically and therefore was used obtain a continuous isotopic and trace element record. *G. bulloides* calcifies in the upper 60 m of the water column and it is assumed that proxy data are representative of a mean calcification depth of 30 m and reflect surface water conditions (Barker and Elderfield, 2002). Accurate determination of magnesium and barium can be biased by preferential dissolution, and/or the

presence of contaminant phases that are mostly attached or adsorbed onto foraminiferal shell tests, such as Fe-Mn oxyhydroxides, Mn-rich carbonate overgrowths, clay minerals and organic detritus (Barker et al., 2003; Boyle, 1983; Hoogakker et al., 2009; Lea and Boyle, 1991). Although the location of GDEC-4-2 in the western Mediterranean has lower salinity waters as compared to sites in the eastern Mediterranean, the impact of general high salinity waters of the Mediterranean Sea (Ferguson et al., 2008)

that can promote secondary carbonate overgrowths was examined by using various chemical cleaning procedures for possible contamination (Fig. S1, S2). The samples were therefore chemically cleaned following the oxidative and reductive-oxidative method to remove any contaminant phases (Boyle and Rosenthal, 1996). We measured Ba/Ca ratios on a ThermoFinnigan Element2 sector field inductively-coupled plasma mass spectrometer (HR-ICP-MS) following the analytical methods detailed by

(Marchitto, 2006). Recurrent analysis of an internal consistency standard solution provides analytical precession of ± 2.6% (1σRSD) for Ba/Ca based on 121 analysis. Reproducibility based on replicate measurements of foraminiferal samples in this study is 5.4% (mean RSD).



Mg/Ca ratios from reductively cleaned samples ("Cd/Ca-cleaning") are identical to oxidatively cleaned samples ("Mg/Ca-cleaning") within instrumental uncertainties (Fig. S1). This rules out the potential problem of Mn-Mg-rich overgrowths generally observed in the eastern Mediterranean due to high salinities (Hoogakker et al., 2009) and preferential dissolution during the reductive cleaning procedure

for accurate paleotemperature estimation from Mg/Ca. The efficiency of the removal of contaminant phases by chemical cleaning was also assessed through the determination of Fe/Ca, Al/Ca and Mn/Ca ratios of the samples. An influence of Fe-Mn oxyhydroxides on Ba/Ca ratios is reflected in a co-variation of Fe/Ca and Mn/Ca ratios with Ba/Ca values. Non-removed clay detritus generally leads to high Al/Ca ratios in the samples. High Mn/Ca ratios may indicate the presence of manganese-rich carbonate

overgrowths (Boyle, 1983; Hoogakker et al., 2009). In sediment core GDEC-4-2, Ba/Ca and Mg/Ca values show no relationship ($r^2$<0.1) with Al/Ca, Mn/Ca and Fe/Ca (Fig. S2). This implies that iron-manganese oxides and clay minerals are not biasing the resultant Mg/Ca and Ba/Ca record.

The range of Ba/Ca and Mg/Ca observed in this study is comparable to the previous studies. For example, the Ba/Ca values in *G. bulloides* calcite is reported to be significantly higher than other planktonic species

collected from core tops across the Mediterranean (Ferguson et al., 2008; Sprovieri et al., 2008). Marr et al., (2013) reported Ba/Ca to range between ~8-14 μmol/mol in *G. bulloides* collected from core tops in southwestern Pacific in the Tasman Sea off New Zealand. Previously, Lea and Boyle, (1991) suggested that several planktonic foraminifera species have high Ba/Ca ratios owing to the differences in the way these foraminifera precipitate their shells. We used McConnell and Thunell, (2005) calibration for SST

determination from *G. bulloides* Mg/Ca measurements, which overlap with our samples in terms of absolute values are applicable for temperatures up to 33°C (Fig. S3).

We applied Mg/Ca-paleothermometry to obtain sea surface temperature (SST) estimates for this region of the Northern Mediterranean (Fig. S1, S2, S3). Because of the absence of coastal upwelling, SST estimates generally should reflect regional atmospheric temperatures, except during times of high riverine

input when water temperatures can be decoupled from local atmospheric conditions. We also develop two independent records for our core site that constrain river runoff: the oxygen isotope ratio of local seawater ($\delta^{18}O_{sw}$) and the Ba/Ca ratio of planktonic foraminifera. Both methods have different sources of uncertainty and are independent of each other. Our working hypothesis is that large changes in the



precipitation and riverine runoff are reflected in the $\delta^{18}O_{sw}$ and budget of dissolved Ba at GDEC-4-2, which are, in turn, archived in foraminifera tests that accumulate in marine sediments.

## 2.3 PMIP3 model simulations

In order to simulate the change in intensity of North Atlantic sourced rainfall in the western
Mediterranean, we analysed mid-Holocene and pre-industrial (PI) simulations from 12 models that participated in the Paleoclimate Modelling Intercomparison Project Phase 3 (PMIP3) (Braconnot et al., 2012), and which are tabulated in Table S1. We only used the latest generation of PMIP experiments (PMIP3; Braconnot et al., 2011; Harrison et al., 2015), which are now a part of the Coupled Model Intercomparison Project phase5 (CMIP5, Taylor et al., 2011). The PMIP3 mid-Holocene (6 ka)
experiment consists of equilibrium simulations run with modified orbital parameters and greenhouse gas concentrations. Mid-Holocene conditions differ from the PI period conditions through their orbital configuration and a reduced concentration of atmospheric concentrations ($CH_4$ reduced to 650 ppb). The orbital configuration modifies the seasonal distribution of heat between the Northern and Southern hemispheres with an increase (reduction) of thermal seasonality in the Northern (Southern) hemisphere.
Aerosols, solar constant, vegetation, ice sheets, topography, and coastlines are prescribed as the same as the PI experiment.

## 2.4 Chronology

We followed the chronology described in Toucanne et al. (2015), which is constrained by aligning the planktonic $\delta^{18}O$, weight percent $CaCO_3$ and XRF-Ca/Ti to the NGRIP ice core $\delta^{18}O$ isotopes from
Greenland for the last glacial termination (GICC05 chronology; Rasmussen et al., 2006; Svensson et al., 2008) and to the synthetic Greenland (GLTsyn) record of Barker et al. (2011) from 60 to ~550 ka BP. For penultimate glacial termination T-II and MIS 5, we synchronized $\delta^{18}O$ of *G. bulloides* to the most up-to-date radiometrically-constrained chronology of ODP Site 975 (Marino et al., 2015) (Fig. 2a; Table S2). Marino et al. (2015) obtained a new radiometrically constrained chronology for ODP975 across T-II and
the last interglacial period to exploit the well-documented intermediate-water connectivity between the eastern and western Mediterranean Sea, and the relationship between marine surface water microfossil



$\delta^{18}O$ and U-series-dated regional $\delta^{18}O$ speleothem records. This was done to obtain a regionally (both eastern and western Mediterranean) synchronous picture for this time period. The $\delta^{18}O$ of planktonic foraminifera *G. bulloides* from the site ODP 975 is synchronized to the Soreq Cave speleothem and $\delta^{18}O$ of *G. bulloides* from marine core LC21 in the eastern Mediterranean, and to $\delta^{18}O$ of *G. bulloides* of ODP

Sites 976, 977 and core MD01-2444 in the western Mediterranean, thereby to the SST and/or IRD records of North Atlantic climate variability that are archived in the Iberian margin sediment cores (see supplementary information, Table S2).

## 3 Results

### 3.1 Proxy systematics

The $\delta^{18}O$ in foraminiferal calcite is controlled by calcification temperature and the $\delta^{18}O_{sw}$. $\delta^{18}O_{sw}$ was estimated using Mg/Ca-based-SST in concert with analysed calcite $\delta^{18}O$ ($\delta^{18}O_{calcite}$) for *G. bulloides* ($\delta^{18}O_{G. bulloides}$) and temperature - ($\delta^{18}O$ calcite - $\delta^{18}O_{sw}$) relationship (Bemis et al., 1998). $\delta^{18}O_{sw}$ in turn is controlled by salinity variations due to river runoff, and changes in the isotopic composition of river water and seawater. The latter in turn reflects changes in precipitation relative to evaporation, advection

of surface waters to the site, and continental ice volume changes.

Foraminiferal Ba/Ca is used as an independent proxy for riverine runoff and rainfall changes (Weldeab et al., 2007). Seawater Ba ($Ba_{sw}$) concentrations at sites influenced by riverine discharge are highly correlated to salinity because dissolved Ba is high in riverine water and Ba desorbs from suspended sediments in estuaries (Coffey et al., 1997). Ba incorporation in foraminiferal calcite varies linearly with

changes in $Ba_{sw}$ and is therefore independent of temperature and alkalinity (Hönisch et al., 2011). We also attempted to use the modern $Ba/Ca_{sw}$-salinity relationship obtained off the Golo River to obtain a first-order estimate of the past runoff-induced SSS variations, as recorded by $Ba/Ca_{foram}$.



## 3.2 Seawater oxygen isotopes and Mg/Ca-based-SSTs

The GDEC-4-2 $\delta^{18}O_{G.\ bulloides}$ record shows high values during glacial periods. The $\delta^{18}O_{G.\ bulloides}$ progressively decrease across Heinrich Stadial (HS) 1 at ~18-15 ka BP and HS-11 at ~135-130 ka BP, and are also characterized by colder Mg/Ca-based-SSTs averaging ~14 and ~13 ºC respectively. The timing of abrupt cooling recorded at our site is consistent with colder SSTs reported for nearby Ocean Drilling Program (ODP) Sites 976 (Martrat et al., 2014) in the western Mediterranean Sea (Fig. 2b, Fig. 3b). The $\delta^{18}O_{sw}$ record also shows concomitant decreasing values, with the largest change (~2.5‰) during HS-11 (Fig. 2c), while Ba/Ca shows no significant change during these periods, suggesting a dominant influence of North Atlantic hydrographic changes in the western Mediterranean and relatively negligible local hydrological changes (inferred from Ba/Ca values) (Fig. 2d). The similarity in magnitude of change in the $\delta^{18}O_{G.\ bulloides}$ at GDEC-4-2 and that observed at site ODP-975 (Marino et al., 2015) (Fig. 2a) indicate that the HS-11 freshwater inputs from deglacial ice-sheet melting entered the Mediterranean Sea via the Strait of Gibraltar impacted ODP 975 and had imprints as far as the Northern Tyrrhenian Sea which is recorded at our site.

The $\delta^{18}O_{G.\ bulloides}$ and calculated $\delta^{18}O_{sw}$ records are characterized by depleted values marking the glacial terminations T-I centred at ~11 ka BP, T-II at ~129 ka BP and T-III at ~243 ka BP with lowest values during the early-mid Holocene MIS 1 (10-6 ka BP), MIS 5e (129-125 ka BP), MIS 7c (220-209 ka BP) and MIS 7e (242-237 ka BP) (Lisiecki and Raymo, 2005), respectively (Fig. 3 a,c). These globally warm periods are characterized by increased Mg/Ca-based-SSTs at GDEC-4-2 with ~18 ºC Holocene values, and MIS 5e being the warmest with temperatures averaging ~24ºC, although at this site, our SST reconstructions indicate increased riverine discharge during MIS 7c and 7e led to local SST being cooler (Fig. S4).

## 3.3 Precipitation and salinity changes inferred from foraminifera Ba/Ca

At GDEC-4-2, high Ba/Ca values characterized the warm periods MIS 1, MIS 5e, MIS 7c and MIS 7e, respectively, indicating enhanced Golo river discharge and lower SSS due to high precipitation over Corsica and by extension in the western Mediterranean. Ba/Ca values are highest during MIS-7c and 7e, when there is a high abundance of benthic foraminiferal assemblages (Toucanne et al., 2015) suggesting





increased import of organic matter to the bottom (Fig. 3d,f). The high Ba/Ca values in *G. bulloides* calcite observed in this study is consistent with previous work using core tops across the Mediterranean and also from southwestern Pacific (Ferguson et al., 2008; Marr et al., 2013). The unique location of core GDEC-4-2 on the upper continental slope at a shallow water depth of 492 m at Golo river mouth in the western Mediterranean limits the occurrence and contribution of barite in Ba measurements, as previously suggested for the western Mediterranean for the Holocene period (Martínez-Ruiz et al., 2003). High Ba/Ca ratios are also accompanied by a corresponding drop in $\delta^{18}O_{sw}$ during each of the warm interglacial periods, MIS 1, MIS 5e and MIS 7c and 7e (Fig. 3f).

## 4 Discussion

### 4.1 Mediterranean hydroclimate changes during the last three interglacials

The last interglacial and the Holocene reconstructed SSTs at our site and other western Mediterranean sites are consistent with that of the eastern Mediterranean, which are much higher than the global average (Rodríguez-Sanz et al., 2017). In the eastern Mediterranean Sea low $\delta^{18}O_{sw}$ values observed at sites LC21 after T-II during the last interglacial (Fig.2g) has been attributed to intensified North African Monsoon as Northern Hemisphere insolation peaked and the ITCZ moved northward, which delivered large amounts of freshwater into the eastern Mediterranean around ~128–122 ka BP (Rodríguez-Sanz et al., 2017). On the contrary, the lower $\delta^{18}O_{G.\ bulloides}$ values observed in the western Mediterranean Sea sites GDEC-4-2 and at ODP 975 (Marino et al., 2015) during the Holocene and the last interglacial, however, reflect the combined changes in global sea level/ice volume, regional evaporation/precipitation and also the warming of surface waters the foraminifera calcified in.

Difficulties in extracting the local evaporation/precipitation signal from the global ice-volume and sea level changes, limits the use of $\delta^{18}O_{sw}$ to examine local hydrological changes. Therefore, at GDEC-4-2 site, the $\delta^{18}O_{sw}$ reflect a combined signature of global ice-volume changes and changes in regional precipitation and the Ba/Ca record for river discharge provide independent estimates of the change in SSS for the past interglacials, with high Golo runoff implying increased precipitation during warm intervals in the western Mediterranean (see Toucanne et al., 2015 and references therein). Accurate estimation of





SSS could not be obtained owing to limited number of paired seawater samples (three paired measurements; Fig. S5) and salinity measurements along the river out to the open sea, nevertheless we have estimated SSS using the available data to get a broad idea of the SSS changes in the past (Fig. 3f). High Ba/Ca implying higher Golo river runoff and lower SSS during the warm interval of MIS 5e is

synchronous with previously published continental and marine records from the surrounding region suggesting high rainfall during this period (Drysdale et al., 2005; Regattieri et al., 2015; Zanchetta et al., 2007). Although the speleothems around the Mediterranean Sea are suggested to have a strong source water effect (Rohling et al., 2015), the periods of lower SSS at our site during MIS 5e correlate with lower $\delta^{18}O_{calcite}$ values in speleothems from Corchia, Tana Urla and the lake carbonates of the Sulmona Basin

in Italy, indicative of high rainfall (Drysdale et al., 2005; Regattieri et al., 2014; Zanchetta et al., 2007) (Fig. 2a, f). Recently, Pasquier et al. (2019) reported episodes of enhanced proportion of land-derived material suggesting significant increase in precipitation amount over the Gulf of Lion catchment area on the Northern Mediterranean borderland during the warm intervals of both Holocene and MIS5, further attesting the regional sedimentary signal in the western Mediterranean. Similarly, palynological evidence

from Greece reveal the expansion of Mediterranean sclerophyllous vegetation and temperate tree pollen, indicative of summer aridity and enhanced winter precipitation during MIS 1 and MIS 5e (Milner et al., 2012; Tzedakis, 2007) (Fig.2e). Together our GDEC4-2 and other discussed proxy records are all for sites which lie outside the influence of the ITCZ-controlled African summer rainfall suggesting enhanced winter rainfall in the entire northern borderlands and western Mediterranean region during the Holocene

and the last interglacial. Interestingly, our geochemical records also show that increased wintertime precipitation and lower SSS in the western Mediterranean extended as far back as the warm intervals of penultimate interglacial, MIS-7c and 7e (Fig. 3f). These results therefore support the hypothesis that high rainfall during interglacials was a distinctive feature of Mediterranean climate (Sierro et al., 2000; Valero et al., 2014), confirming that the precession minima (boreal summer insolation maxima and winter

minima) paced rainfall variability.

The most striking feature in our geochemical record is a change in the amplitude of Ba/Ca (and Ba/Ca-derived SSS, Fig. 3f) that follows the eccentricity cycle, with higher Ba/Ca (and lower SSS) during MIS 7, relatively lower Ba/Ca (and higher SSS) in MIS 5, and the lowest Ba/Ca (and highest SSS) in MIS 1,





corresponding to the respective insolation maxima of the 100 ka-eccentricity cycle during each interglacial (Fig. 3f, g Fig. S6). This trend in the change in magnitude of river discharge due to winter rainfall confirms the idea of eccentricity modulation of the precession-driven rainfall/runoff in the Mediterranean during the late Pleistocene, as has been hypothesised previously for the Pliocene (Sierro et al., 2000).

## 4.2 Proposed mechanism for high Mediterranean winter precipitation during interglacials

We used climate model simulations from the Paleoclimate Model Intercomparison Project – Phase 3 (PMIP3)(Braconnot et al., 2011) to shed light on the variability of winter precipitation during these times and also to examine the source of the wintertime Mediterranean rainfall. Our model analysis for a representative interglacial (the mid-Holocene at ~6 ka (as used in PMIP3) compared to pre-industrial (PI) conditions) suggests enhanced southwesterly mean moisture transport from the North Atlantic causing higher moisture convergence during winters in the Mediterranean, potentially brought about by a south-eastward shift of storm tracks (Fig. 4a) during interglacials, in a negative North Atlantic Oscillation (NAO)-type pattern. Recent extreme rainfall events over the northern Mediterranean borderlands have a distinct North Atlantic origin of moisture (Celle-Jeanton et al., 2001). Today, NAO is the dominant atmospheric phenomena in the North Atlantic region (Olsen et al., 2012), including the northern Mediterranean borderland (Hurrell, 1995; Trigo et al., 2002) (Fig. 1), such that during the negative phase of NAO, storm tracks are shifted southwards that bring wet and mild winters over the southern Europe. Fluctuations in NAO strongly affects the intensity of zonal flows over the North Atlantic (i.e. westerlies), the position of storm tracks and subsequent precipitation amount across Europe and the Mediterranean basin (López-Moreno et al., 2011). Recent coupled atmosphere–ocean general circulation model suggest that these NAO-type mode of climate variability could also have operated at orbital timescales such as MIS 5e (Lohmann, 2017). A North Atlantic connection of winter rainfall is also suggested previously using palynological proxies from the Iberian margin (Amore et al., 2012) and geochemical proxies from the Gulf of Lion (Pasquier et al., 2019). Furthermore, all the mid-Holocene model outputs from our model analysis are in good agreement with the mid-Holocene high lake levels, which indicate increased precipitation minus evaporation (P – E) due to increased precipitation during the early-mid Holocene (10-



6 ka BP) on the northern Mediterranean borderlands (Magny et al., 2013) (Fig. 4b). A similar pattern of wetter winter with a strong seasonal cycle of surface air temperatures during the early Holocene was also observed in previous general circulation model simulations (Brayshaw et al., 2011). In particular, a stronger southwesterly flow during the winter 6kaBP experiment (compared with the PI control run) was clearly shown such that the northern coast and western Mediterranean received strong precipitation (Brayshaw et al., 2011). Comparison of Holocene proxy-models using regional scale downscaling of a set of global climate model simulations for the Mediterranean region also give consistent results (Peyron et al., 2017).

Additionally, there is also evidence for stronger seasonality in winter precipitation and P − E during interglacials in the PMIP3 simulations (Fig. S7), due to an intensifying moisture convergence in late winter, as previously suggested by palynological records from Greece and Turkey (Milner et al., 2012; Tzedakis, 2007). Previous modelling experiments demonstrate increased winter precipitation in the regions between 30 ºN and 45 ºN over the Mediterranean during periods of maximum orbitally forced-seasonality (Kutzbach et al., 2014). Together, these simulations point to wetter winters in the northern Mediterranean borderlands with a stronger seasonal cycle of surface air temperatures during the early Holocene.

There is ample evidence suggesting that North African precipitation was at a maximum during the mid-Holocene and during other interglacials (Ziegler et al., 2010; Rohling et al, 2015 for a complete review). Maximum Northern Hemisphere seasonality (summer perihelion–increased insolation; winter aphelion–decreased insolation) has been linked to intensified summer monsoon rainfall over North Africa and also increased Mediterranean storm tracks precipitation in winters (Kutzbach et al., 2014). The analysis of PMIP3 simulations carried out in this study also demonstrate intensified African summer monsoon rainfall through the mid-Holocene, during times of enhanced winter precipitation (Fig. S8).

## 4.3 Contribution of western Mediterranean precipitation in sapropel deposition

Toucanne et al., (2015) used sedimentological proxies to suggest that increased winter storm tracks into the western Mediterranean basin during the past interglacials were sourced from the North Atlantic and contributed to basin-wide anoxia. In contrast, Rohling et al., (2015) refuted any external moisture source





in the Mediterranean and purported that any increase in moisture in the Mediterranean was in fact recycled moisture sourced from the Mediterranean itself and, therefore, had little effect on the overall hydrological budget of the basin. Our direct geochemical evidence, coupled with model simulations, possibly suggest a direct addition of extra-Mediterranean (i.e. Atlantic) water into the Mediterranean Sea during the warm

past interglacials and during periods of northward position of the ITCZ. The close correspondence of the timing of intensified African summer monsoon rainfall and North Atlantic-sourced winter rainfall, as indicated by both proxy reconstructions and model analysis, suggest a close coupling between low and high latitude atmospheric-oceanic processes in triggering anoxia in the Mediterranean. The timing of increased rainfall during MIS 1, MIS 5e, MIS 7c and MIS 7e coincides with anoxic events in the

Mediterranean that caused the deposition of organic-rich sapropels S1, S5, S8 and S9, respectively (Grant et al., 2016; Ziegler et al., 2010). Such mid-latitude storm tracks originating from the North Atlantic contributed to increased organic fluxes into the Mediterranean Sea, maintained the already-disrupted hydrology of the Mediterranean, and reduced the intermediate and deep-water ventilation. Indeed, the oceanic response to increase freshwater flux is a reduction in salinity and mixed layer depth that lead to

stronger stratification and less Western Mediterranean Deep Water formation (i.e. WMDW; Meijer and Tuenter, 2007; Rohling et al., 2015 and reference therein). Our study provides new geochemical constraints to the previously suggested hypothesis that the penetration of direct Atlantic moisture into the Western Mediterranean basin caused North Mediterranean borderland's cyclogenesis and rainfall (Drysdale et al., 2005; Kallel et al., 2000; Reale et al., 2001; Regattieri et al., 2014, 2015; Rohling and

Hilgen, 1991; Toucanne et al., 2015; Zanchetta et al., 2007). While these processes likely would have worked in concert with warming of the Mediterranean Sea to have triggered anoxia, the exact role of Atlantic origin freshwater during periods of sapropel deposition need further investigation.

## 5 Conclusions

In this study, we used geochemical proxies to better assess the variation in winter rainfall during the

Holocene and the past two interglacials. Our geochemical data and model analysis support variability in the contribution of North Atlantic- sourced winter rain to the Mediterranean during the last three interglacials. Proxy data placed on a globally synchronous timescale demonstrate that the intensity of the



precession-controlled wintertime rainfall in the western Mediterranean was modulated by eccentricity, with times of high eccentricity characterised by higher rainfall and river outflow. These results along with the analysis of Holocene climate simulations support increased winter precipitation sourced from the North-Atlantic in a warmer western Mediterranean during the past. Data and model analysis support the

hypothesis that northern borderlands were a significant source of freshwater into the Mediterranean basin during past interglacials. Comparison of our record with other proxy reconstructions from the northern Mediterranean borderlands reveals the regional character of these wet periods. Our data and model results also show that high rainfall events in the Northern Mediterranean borderland occurred at time of intensified North African summer monsoon and the sapropel deposition in the Mediterranean basin. The

close chronological correspondence of increased river outflow and winter rainfall to organic carbon deposition and sapropel occurrence supports a causal link. We suggest a close coupling between low and high latitude atmospheric-oceanic processes in triggering anoxia in the basin, with a role for both Nile River outflow changes due to variations in African summer monsoon rainfall as well as North Atlantic climatically-controlled winter-rainfall driving outflow changes in the western Mediterranean.

## Acknowledgements

YD, ST and AT thank the "Laboratoire d'Excellence" LabexMER (ANR-10-LABX-19) and the French government ("Investissements d'Avenir") for support. YD was supported by a grant from the Regional Council of Brittany (SAD programme), and by the EU FP7 Marie Curie actions (PCOFUND-GA-2013-

609102), the PRESTIGE programme (Campus France). AT and JL acknowledge support from a NSF CAREER award (NSF EAR-1352212), and JL was also supported by a NSF AGS postdoctoral fellowship, a Chancellor's postdoctoral fellowship, and a California Alliance Postdoctoral Fellowship. We thank B. Gougeon, M. Guillermic, Y. Germain and M. Greaves (University of Cambridge) for help with the trace element analysis and Y. Baldi for collection of seawater. The authors also thank the crew

and scientific members of the GOLODRILL cruise (R/V 'Bavenit' - FUGRO) for the recovery of the GDEC-4-2 borehole in the frame of the "GOLO PROGRAM", a research consortium between IFREMER, TOTAL, EXXONMOBIL and FUGRO. We acknowledge the World Climate Research Programme's Working Group on Coupled Modelling, which is responsible for CMIP, and thank the climate modeling





groups (listed in Table S1 of this paper) for producing and making available their model output. For CMIP, the U.SS. Department of Energy's Program for Climate Model Diagnosis and Intercomparison provide coordinating support and led development of software infrastructure in partnership with the Global Organization for Earth System Science Portals.

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



## Figures

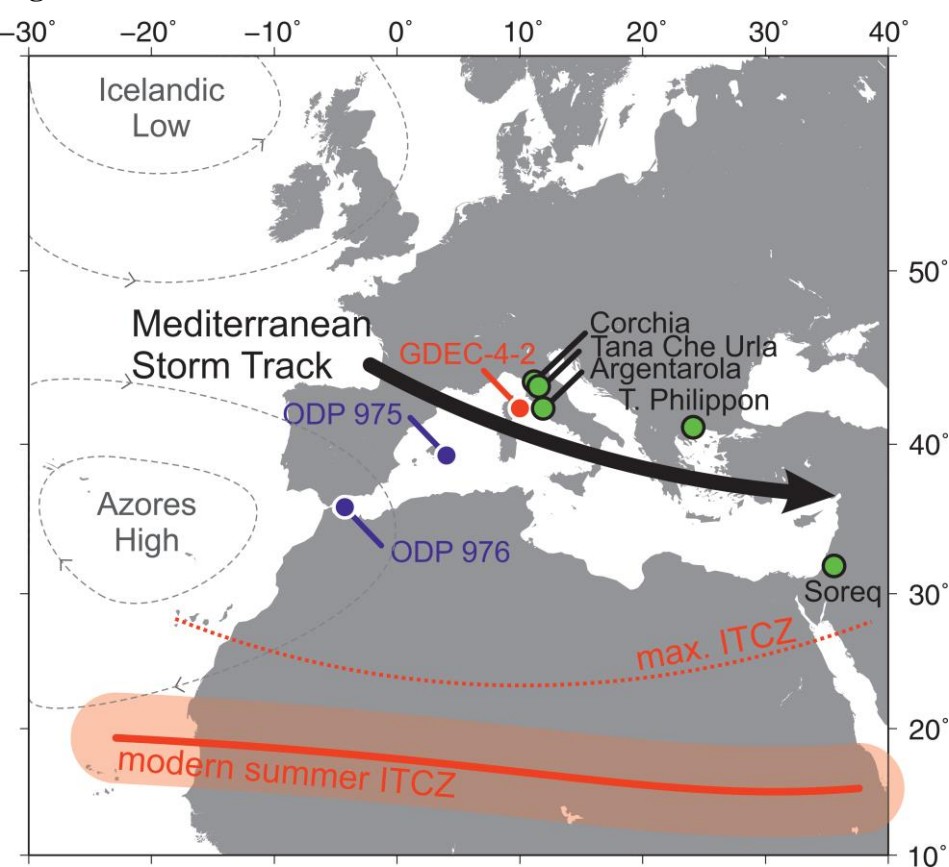

**Figure 1: Location of GDEC-4-2 (red) in the Northern Tyrrhenian Sea and other marine (blue) and terrestrial archives (green). Red band and red dotted line denotes the extent of modern summer ITCZ and the maximum northward reach of ITCZ in the past respectively. Also shown are the sea-level pressures in North Atlantic and the direction of Mediterranean storm tracks (black).**

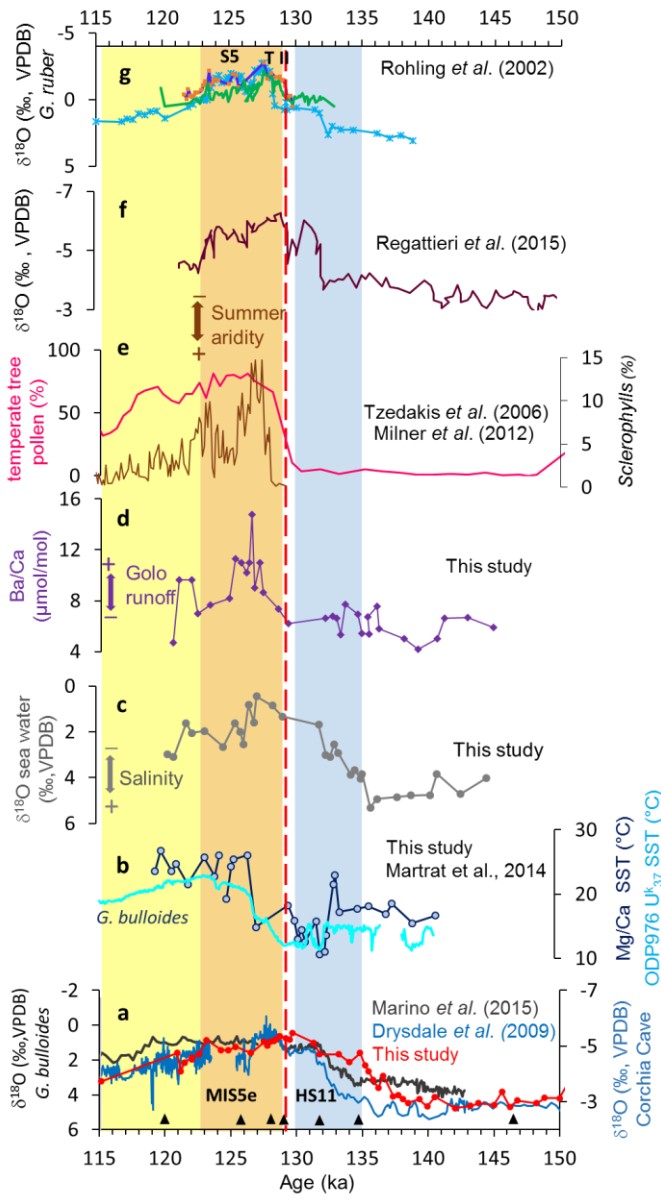

**Figure 2: Comparison of GDEC-4-2 results with regional records for the last MIS 5e. (a)** $\delta^{18}O$ *G. bulloides* **of GDEC-4-2 (red), ODP 975 (grey) (Marino et al., 2015) and** $\delta^{18}O$ **of Corchia Cave speleothem (blue) (Drysdale et al., 2005) (b) Mg/Ca-based SSTs from *G. bulloides* (blue) from GDEC-4-2 and UK37- SSTs from ODP 976 (light blue)(Martrat et al., 2014); (c)** $\delta^{18}O$ **of sea water (uncorrected for global ice volume changes) obtained using** $\delta^{18}O$ *G. bulloides* **and Mg/Ca-based SSTs from *G. bulloides* from GDEC-4-2; (d) Ba/Ca in foraminifera as a proxy of river discharge in GDEC-4-2; (e) Pollen data: temperate pollen (pink) and Sclerophyllous (brown)(Milner et al., 2012; Tzedakis, 2007); (f)** $\delta^{18}O$ **of Tana urla Cave, Italy (Regattieri et al., 2014); (g)** $\delta^{18}O$ **of *G. ruber* from sapropel cores in eastern Mediterranean (Rohling et al., 2002). Vertical orange and light yellow bars denote warmer substages of the last interglacial respectively. Heinrich stadials (blue bar) and glacial termination (dashed red line) shown on top. Blue bar denotes Heinrich stadial HS11, based on Marino et al., (2015) chronology. Also shown at the bottom are tie points (black triangles) to synchronize GDEC-4-2 with the most up-to-date chronology of ODP 975.**



**Figure 3. Comparison of records for current interglacial (Holocene) and last interglacial and penultimate interglacial from core GDEC-4-2 to other datasets. (a)** $\delta^{18}O$ *G. bulloides* **(red); (b) Mg/Ca-SST (this work; blue) and Uk37–SST (ODP 976; light blue) (Martrat et al., 2014); (c)** $\delta^{18}O_{sw}$ **(grey) (uncorrected for ice-volume changes (this work); (d) Deep infaunal foraminifera abundance (Toucanne et al., 2015); (e)** *G. bulloides* **Ba/Ca (analytical precession of ± 2.6%) (this work; purple) and Ba/Ca-based SSS estimates and eccentricity (black dashed line); (f) Precession and June insolation at 45 °N. Vertical orange and light yellow bars denote warmer substages and stadials of interglacials respectively. Sapropel deposition intervals, Heinrich stadials (blue bar) and glacial terminations (dashed red line) shown on top.**

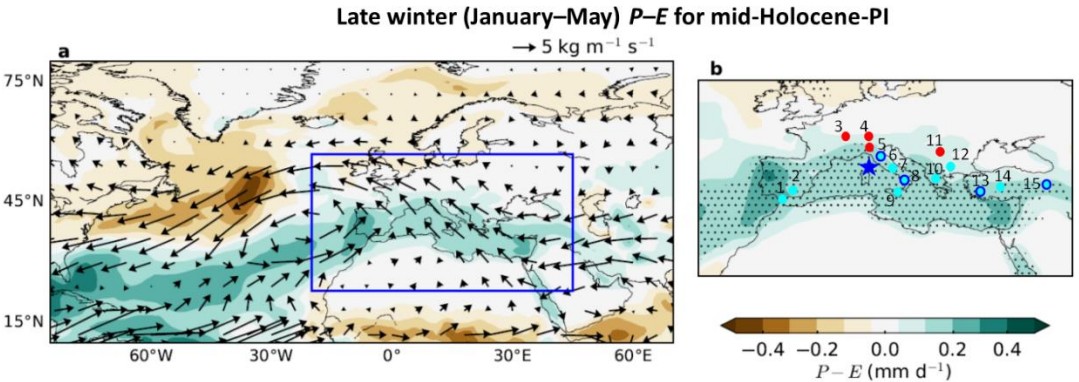

**Figure 4. a) The change in late-winter (January–May) P–E between mid-Holocene and pre-industrial simulations (multi-model ensemble for 11 PMIP3 models) is shown using colored contours. A band of higher Holocene moisture convergence runs from the southwestern North Atlantic to the Iberian Peninsula, and across the Mediterranean, accompanied by a decrease in the north and northwestern North Atlantic. The corresponding differences in mean moisture transport for the region (arrows) indicate increased southwesterly transport into the western Mediterranean, as well as decreased westerly transport (shown by easterly arrows) across the eastern Mediterranean as well as the northern North Atlantic. b) Zoomed map of the boxed region in (a), showing the same P–E difference in the Mediterranean, with stippling showing where at least 9 of the 11 models agree on the sign of the change. The numbers denote sites with data constraining hydroclimate changes (this work- blue star, high rainfall/ high lake levels- blue circles, dark blue lined for past three interglacials) that are consistent with high winter precipitation during the early-middle Holocene. Red circles denote the lakes with lower lake levels during the Holocene. See supplementary information for details of each record.**