# Peer review of "Enhanced western Mediterranean rainfall during past interglacials driven by North Atlantic pressure changes"

_Climate of the Past, 2019_

## Referee Comment (RC1) · Anonymous Referee #1 · 1 Aug 2019

The paper from Dixit et al present trace elements (Ba/Ca and Mg/Ca) and stable oxygen isotope composition from planktic foraminifera (G. bulloides) from the previously published marine core GDEC-4-2 from the Corsica margin. The data cover the Holocene, the interglacials MIS5 and MIS7 and the glacial termination TI, TII, TIII. Sea surface temperature were obtained from Mg/Ca data and used together with $\delta$18O calcite data to calculate $\delta$18O of sea water. Ba/Ca is used as a proxy for Golo River discharge and, through calibration using the modern sea surface salinity (SSS)-Ba/Ca relationship, the authors attempt to quantitatively reconstruct past SSS. Using these data, and by comparison with several records (both marine and continental) from the Mediterranean, the authors suggest that the three interglacial were characterized by

an increase in winter precipitation driven by changes in North Atlantic storm tracks trajectories, in turn modulated by changes in precession and eccentricity. To support their hypothesis, the authors also analysed outputs from modelling experiments (PMIP3) for the pre-industrial Holocene. Authors also suggests that these increases in precipitation contributed to trigger basin anoxia and sapropel deposition. The presented data are of interest and their interpretation as paleo-rainfall variability proxies is reasonable. However, main text, figures and supplementary material are rather confused and misleading in some points, references are not updated and often messed-up, the comparison with the model almost useless and the whole discussion is a bit inconsistent and not fully supported by the data. Moreover, the main findings of the paper add very few to what was already proposed in the original paper on the same core (Toucanne et al., 2015). Thus, I suggest publication in Climate of the Past only after careful major revision.

Main Points: One of the main claims of the paper is that variations in winter precipitation are modulated by eccentricity changes. I found this claim a bit obvious. Indeed, it is well known from both data and modelling experiments that winter precipitation in the Mediterranean are mostly modulated by precession changes (e.g. Tzedakis et al., 2007; Milner et al., 2012; Toucanne et al., 2015; Regattieri et al., 2015; Bosmans et al., 2015, just to quote some, but there are others). The intensity of the precession forcing relate to changes in eccentricity, with higher eccentricity inducing higher precession forcing and lower eccentricity reducing the influence of this orbital parameter. Honestly, I do not see any reason to decouple the forcing related to precession to that related to eccentricity. Also, the importance of obliquity changes is not mentioned at all, while several works (se e.g. Bosmans et al., 2015 and references therein) have showed that it has an impact on the Mediterranean hydroclimate. I think that authors have to largely re-focus the discussion, better explaining the relationship between eccentricity and precession and taking into account the influence of obliquity changes. To this end, I suggest that they have a detailed look to Bosmans et al (2015) results. Another very weak point is the claim that the findings of the paper are supported by modelling results. I found this part confused and even misleading. First because the

model output is related only to the mid-Holocene, so results cannot be extended to others interglacials where the boundary conditions were so different, second because the whole discussion about changes in storm tracks trajectories and NAO like atmospheric patterns are not supported, to me, neither by the data or by the modelling experiment that they present. I can agree that the Mid-Holocene experiments shows a southern shift in westerly trajectories which can resemble a NAO- pattern, but I do not see any reason to extend this interpretation to the other considered periods. The most likely mechanism for increased precipitation during precession minima, basing on available data and literature, is related to changes in Mediterranean-sourced precipitation due to increased Med heat content. Indeed, during precession minima hydrology in the Med is influenced by low-latitudes atmospheric patterns: the northward shift of the ITCZ causes stronger summer drought related to the descendent branch of the Hadley cell. It causes an increase in the Mediterranean summer heat content. High summer temperatures lead to elevated sea-surface temperatures and associated high evaporation levels persisting well into the year, contributing to the formation of depressions across the northern borderlands, strengthening cyclogenesis within the basin and causing an increase in autumn-winter precipitation. This is what has been proposed also basing on GDEC data by Toucanne et al just few years ago, and I do not understand why authors now invoke a completely different mechanism. . ... The whole discussion about the model experiment is very confused, do not add nothing to the interpretation and do not support what the paper claims. I suggest to largely modify section 4.2 trying to explain the mechanisms more relying on the presented data and on previous literature. They should briefly review mechanisms proposed by e.g. Tzedakis et al. (2007) or by Milner et al. (2012) or by Bosmans et al. (2015) and especially by Toucanne et al., 2015, trying to better highlight which one best fits with their results. To me, this whole part about modelling is an, almost failed, attempt to add something new to the -good- explanations already proposed by Toucanne et al. Authors should be more "honest" with that in the sense that they should clearly state that this work is an update of the previous one and that the new data and results strengthen the previous interpretation, without striving

to introduce new and confused mechanisms for that. Specific points: Abstract: P. 1 line 22: North Atlantic climatic processes is rather vague, do the authors refer to atmospheric patterns? or to oceanic circulation? P1 line 23: (but also elsewhere, see above and below) Summer monsoon rainfall does not reach directly the Mediterranean Basin. As this sentence stands now, it seems that monsoon directly contribute to Mediterranean precipitation. I agree that monsoon rain contribute to Mediterranean Sea water through Nile (and fossil river system from N Africa) discharge, but it has to be clearly explained. p1 line 25 across the past 1-Introduction: P2 line 16 Hydrological not hydrologic p3 line 1 There is a typo in interglacials p3-line 3-9 this part reads odd. Please rephrase. I guess the words between "Mediterranean" and" for" should be moved after "(Railsback et al., 2015)", also it is not clear which papers refer to Holocene and which to the LIG (e.g. Zanchetta et al., 2007 is Holocene, not LIG). p3- line 13 because THE of a lack p3 line 20 to the end of the section: it should be moved in a paragraph of site description or in material and methods, it is not introduction. IMPORTANT: a sentence clearly explaining the aim of the paper is missing from the introduction, please add it at the end.

2 Material and methods p4 line 14 GDEC is WAS RECOVERED from P4 line 18-20 Rephrase, a sediment core cannot capture variation in storm track (sediment properties yes, but it should be better explained). 2.1 Stable isotope analyses Which was the previous resolution of stable isotope analyses? which is the new one? There are not enough details about analytical method (i.e. which calibration method has been used?, which is the reaction time? If analytical methods were the same as in the Toucanne et al paper, it should be stated clearly. 2.2 Trace elements analyses Add the resolution (spatial) at which these analyses were done. p.5 line 12 proxy data OBTAINED FROM IT are representative... p5 line 13 30 m, and are reflect THUS REFLECTING surface... p5 line 25 TO CHECK FOR INTERNAL CONSISTENCY recurrent analyses... p5 line 26 precession PRECISION p6 line 13 "comparable" to the TO THAT OBSERVED IN previous studies p6 line 15 Here it is stated that others core top in the Mediterranean have lower Ba/Ca values so it is not clear if Ba/Ca values observed here are compa-

rable or not with previous studies. . .. Also the sentence about calibration for used to infer temperature from Mg/Ca (line19-21) should be better separated by the discussion about Ba/Ca and better motivated. It is the same calibration used in previous studies in the region or not? (i.e. from where the McConnell calibration comes from?) p6 line 22-23 for this region of the Mediterranean P6 line 24 generally p6 line 26 records TO for our core site that constrain river runoff AT OUR SITE 2.3 PMIP3 model simulation See general comments, it is a non-sense to use a mid-Holocene simulations to infer mechanisms working for other interglacials characterised by different boundary conditions. I would almost remove this part. . .please instead consider modelling results from Bosmans et al. 2015 paper. 2.4 Chronology As the GDEC record has been published already and now the chronology is updated by aligning to the Marino et al. (2015) curve I suggest the authors to quantify the difference with the previously published record. Also, associated uncertainty to the new chronology has to be stated clearly. Last, at the end of this paragraph authors should insert the resulting temporal resolution for both the stable isotope and the trace element records. p7 line 25: to exploit EXPLOITING

3 Results 3.1 Proxy systematics p.8 line 10 "$\delta$18O OF in foraminifera" and "and BY $\delta$18Osw". Also, you should put a reference here and also quote Fig. S3. p.8 line 14. Sentence not clear. Also the $\delta$18O of the river water is related to P/E ratio, not only the $\delta$18O of the sea water. At the end of the page you should quote the relative supplementary text and figure. 3.2 Sea Water oxygen isotope and Mg/Ca based SSTs p.9 line 2 highER, not high and also quote a figure after periods p.9 line 4 and THESE INTERVALS ARE also characterized p.9 line 5-6 This is not results but discussion already (see also comments to Figure) p.9 line 16 BP, with AND BY lowest values Authors should briefly comment here about MIS7 temperatures and removing the relative paragraph, which is really confused, from the supplementary material (see specific comments to Supp Mat). 3.3 Precipitation and salinity changes inferred from foraminifera Ba/Ca p.9 line 27 Why the increase abundance of benthic foraminifera indicates an increase of OM transportation to the bottom? 4 Discussion p10 line 11 Last Interglacial (here and after) p10 line 15 which delivered DELIVERING p.10 line 16 As above, you should

specify that the monsoonal rain is delivered by the Nile and by -now fossil- river system in the North Africa p10 line 20 waters FROM WHICH the foraminifera calcite FORMS p10 line 25 increased LOCAL precipitation p11 line 4 put a comma after Ba/Ca and another one after MIS5e p11 line 5 synchronous TO WETTER CONDITIONS INFERRED p11 line 7 as above, Zanchetta et al., 2007 is Holocene and not LIG, I guess Regattieri et al., is 2014 or 2017 and not 2015. Lines 9-10 are a repetition of lines 5-6. p11 line 8 The source effect on continental calcite in the Mediterranean has suggested to be strong during glacial to interglacial transition, but less important during interglacial period (see e.g. Tzedakis et al., 2018 or Regattieri et al., 2019) p11 line 14 What does "regional sedimentary signal means"??? p11 line 17 Tzedakis et al, 2007 does not report any Holocene pollen record showing higher seasonality and for should be FROM sites In general, what is new in this paragraph with respect to the Toucanne et al paper???? The whole discussion about eccentricity influence is a non-sense, as the amplitude of the precession forcing is directly related to eccentricity!

4.2 Proposed mechanism for high Mediterranean winter precipitation during interglacial See general comment. I would largely remove this part. . .

4.3 Contribution of western Mediterranean precipitation in sapropel deposition (in should be TO instead of in) Toucanne et al paper's speaks about an increase of western Mediterranean storm track, not about an increase in North Atlantic sourced precipitation during period of sapropel deposition. I agree that wMed precipitation play a role in triggering anoxia and sapropel deposition and I do not support as well the Rohling hypothesis. However, this part is very confused and I do not see any reason to invoke an increase of moisture transport from the North Atlantic. This claim is not supported by the references provided in lines 19-20, nor by the new presented data, and is in contrast with what already proposed basing on GDEC data. p14 line 1 there's a typo in supported (or proposed?) p14 line 12 how mid-latitude storm tracks can contribute to organic fluxes? this sentence has no sense. Conclusion: they need to be largely rewritten following provided comments. Figures They are all rather poorly constructed

in my opinion and need to be largely modified. I suggest to prepare a proper results figure showing only the results from GDEC for all the period discussed (this should be fig. 2 not 3), then to make others figures with the three intervals separated and where the records used for comparison have to be shown. Please enlarge all the figure and be sure that axes's values are appropriated. Figure 4 is useless in my opinion, all the mentioned sites needs to be shown in fig. 1 If you want to show Corchia data please use the updated record provided by Tzedakis et al., 2018 Fig. 1 The line indicating the Mediterranean storm tracks has no sense, this line may resemble the major trajectory of North Atlantic storm track, but it seems to me an over simplification. Argentarola cave is not mentioned in the text, why it is mentioned here? From where the position of the ITCZ comes from? again it seems poor and over simplified. Please put all the reference for the terrestrial and marine sites in the caption of Figure 1, this would avoid the whole first paragraph of supplementary text, which is really confused and not useful at all. Fig.2 It should report only the results from GDEC, whereas all the other records used for comparison should be moved to another figure (fig.3)

Supplementary information The first two paragraph (regarding the records used for comparison and the one regarding the MIS7 temperature, should be shorten and accommodated in the main text and in figure captions as indicated in previous comments. Fig. S5: Why there are only 3 points if in table s5 five sampling points are reported? The high correlation coefficient reported is simply an artefact due to the very limited number of points!

References: Milner, A. M., Collier, R. E., Roucoux, K. H., Müller, U. C., Pross, J., Kalaitzidis, S., ... & Tzedakis, P. C. (2012). Enhanced seasonality of precipitation in the Mediterranean during the early part of the Last Interglacial. Geology, 40(10), 919-922. Bosmans, J. H. C., Drijfhout, S. S., Tuenter, E., Hilgen, F. J., Lourens, L. J., & Rohling, E. J. (2015). Precession and obliquity forcing of the freshwater budget over the Mediterranean. Quaternary Science Reviews, 123, 16-30. Regattieri, E., Giaccio, B., Zanchetta, G., Drysdale, R. N., Galli, P., Nomade, S., ... & Wulf, S. (2015). Hydrological variability over the Apennines during the Early Last Glacial precession minimum, as revealed by a stable isotope record from Sulmona basin, Central Italy. Journal of Quaternary Science, 30(1), 19-31. Regattieri E., Isola I., Zanchetta, G., Tognarelli, A., Hellstrom, J.C., Drysdale R.N., Boschi, C., Milevski, I., Temovski, M. 2019. Middle-Holocene climate variability from a stalagmite from Alilica Cave (Southern Balkans). Alpine and Mediterranean Quaternary, 32 (1), DOI: 10.26382/AMQ.2019.02 Regattieri, E., Zanchetta, G., Drysdale, R. N., Isola, I., Hellstrom, J. C., & Roncioni, A. (2014). A continuous stable isotope record from the penultimate glacial maximum to the Last Interglacial (159–121 ka) from Tana Che Urla Cave (Apuan Alps, central Italy). Quaternary Research, 82(2), 450-461. Regattieri, E., Giaccio, B., Nomade, S., Francke, A., Vogel, H., Drysdale, R. N., ... & Boschi, C. (2017). A Last Interglacial record of environmental changes from the Sulmona Basin (central Italy). Palaeogeography, Palaeoclimatology, Palaeoecology, 472, 51-66. Toucanne, S., Minto'o, C. M. A., Fontanier, C., Bassetti, M. A., Jorry, S. J., & Jouet, G. (2015). Tracking rainfall in the northern Mediterranean borderlands during sapropel deposition. Quaternary Science Reviews, 129, 178-195. Tzedakis, P. C. (2007). Seven ambiguities in the Mediterranean palaeoenvironmental narrative. Quaternary Science Reviews, 26(17-18), 2042-2066. Tzedakis, P. C., Drysdale, R. N., Margari, V., Skinner, L. C., Menviel, L., Rhodes, R. H., ... & Fallick, A. E. (2018). Enhanced climate instability in the North Atlantic and southern Europe during the Last Interglacial. Nature communications, 9(1), 4235.

---

## Referee Comment (RC2) · Anonymous Referee #2 · 29 Aug 2019

The manuscript submitted by Dixit et al to the Climate of the Past journal is a revised version of a manuscript that I reviewed for another journal almost two years ago, and I must say that several of my concerns have not been suitably addressed. Dixit et al. present Mg/Ca and Ba/Ca-based reconstructions of sea surface temperature (SST) and salinity, respectively, for the Tyrrhenian Sea during the last three terminations (TI, TII en TIII) and peak interglacials of MIS 1, MIS 5e, MIS 7e and MIS 7c. From these reconstructions the authors infer changes in Golo River runoff that they interpret to indicate changes in winter rainfall over the northern Mediterranean basin. They observe that the long-term amplitude of the salinity decrease tightly follows eccentricity. They also find that during SST warming putative increases in winter rainfall

coincide with increases of African monsoon, increase of Nile River runoff in summer, both developing well-stratified column waters, periods of anoxia and sapropel layers. A comparison of these results with model simulations for the mid-Holocene allows the authors to support the idea of an increased southwesterly moisture transport into the western Mediterranean from the North Atlantic. The first observation is new, and it is interesting to know the factor that may modulate the amplitude of long-term changes in salinity in the Mediterranean Sea; the second finding is not new but supports previous studies explaining sapropel formation (Toucanne et al., 2015, QSR; Grant et al., 2016, QSR) and the origin of the western Mediterranean rainfall during the mid-Holocene (Brayshaw et al., 2011, The Holocene). Therefore, I do not see the real contribution of this manuscript. Moreover, interpreting changes in runoff as direct evidence of changes in seasonal (winter) rainfall seems to me to be inappropriate. Changes in runoff can be the result of changes in vegetation cover with increased runoff during late summer/early autumn in the Mediterranean associated with more erosion, i.e. less forest cover (Durán Zuazo and Rodríguez Pleguezuelo, 2008, Agronomy for Sustainable Development). Additionally, changes in salinity can also result from changes in precipitation-evaporation balance. These issues are not sufficiently discussed in the manuscript. Based on previous research in the Mediterranean region, they state that other proxies (pollen and speleothems) are "unable to offer direct insights on the variability in winter rainfall". Contrary to this statement and as far as pollen studies are concerned, we know that present-day changes in Mediterranean forest cover depend on the North Atlantic Oscillation shifts, i.e. on the position and intensity of the westerlies, that in turn control winter precipitations in Europe (Gouveia et al., 2008, Int. J. of Climatology). Therefore, pollen-based Mediterranean forest cover changes are direct evidence of changes in winter precipitation as repeatedly shown by data (Fletcher and Sanchez Goñi, 2008, Quat. Res.) and model–data comparisons for different interglacials of the last 800,000 years (Peyron et al., 2017, Climate of the Past; Oliveira et al., 2018, Climate Dynamics). Moreover, some of these records have allowed for quantitative reconstructions of winter precipitation for TI and the peak of MIS 1 (Fletcher et

al., 2010, Climate of the Past; Peyron et al., 2017). I am surprised by the fact that the authors refer to some of these papers in the Discussion section to support their interpretation after criticizing such an approach in the Introduction. Moreover, they justify their work by the inability of this proxy to reconstruct winter rainfall. Throughout the manuscript the authors are not consistent when they refer to the region of precipitation. Sometimes they refer to northern Mediterranean rainfall, at other times to western Mediterranean rainfall, and they discuss records coming from the east and to the west of this region. This inconsistency is problematic as several studies show that climate during the Holocene in the Mediterranean region presents west-east and north-south gradients (e.g. Dormoy et al., 2009, Climate of the Past). I am also concerned by how the authors deal with the timing of Terminations. Terminations are intervals from glacial to interglacial states that generally last a few thousand years and are not events (midpoints) as suggested by Dixit et al. (dashed line in Figures 2 and 3). Terminations should be identified from the $\delta$18O of benthic foraminifera, and they are triggered by a combination of ice volume and orbital parameters (Parrenin and Paillard, 2012, Climate of the Past). Cheng et al. (2009, Science) established the timing of marine oxygen-isotope terminations ($\delta$18O of benthic foraminifera) by correlating North Atlantic ice rafted debris (IRD) to radiometrically dated oxygen-isotope cave records from China. The timings of the onset and end of Terminations I, II and III are 18-11 ka, 138-129 ka, 251-243 ka, respectively, and the timing of the midpoint terminations are 14.5 ka, 131 ka, 247 ka. These accurate measurements of the timing of terminations do not coincide with the dates given by Dixit et al. The authors say that the three terminations are centered on 11, 129 and 243 ka, but they do not specify how they established them. With respect to this issue, I invite the authors to look at the recent paper by Barker et al. 2019 in Paleoceanography and Paleoclimatology. Overall, the organization of the manuscript and the order of the figures should be changed, and the English improved. For instance, the environmental setting and the studied material are explained twice: at the end of the introduction and at the beginning of the Material and Methods. The introduction should be more focused and clearly explain the gap this work aims to fill and

justify the interest of working and comparing the MIS 1, MIS 5e and MIS 7e and MIS 7c warm periods and the related Terminations. Adding model simulations of precipitation changes during contrasting MIS 5e, MIS 7c and MIS 7d interglacials could be relevant. The subsection "Proxy systematics" should be moved to the Material and Methods section. Furthermore, I do not understand the meaning of "systematics" in this context. Figure 3 in which all the results from this study are shown should be Figure 2. Figure 2 is only displaying different records covering TII and MIS 5e in the Mediterranean. Additionally, I have found many inconsistencies throughout the manuscript, sentences difficult to understand, and several typographic mistakes (see below other comments). In the conclusion section I have one major concern related to the following sentence: "Proxy data placed on a globally synchronous timescale demonstrate that the intensity of the precession-controlled wintertime rainfall...": What do the authors mean by "globally synchronous timescale"? How have the authors harmonized the different paleoclimatic records presented in the work: GDEC-4-2, ODP sites 975 and 976, Corchia and Tana Urla speleothems, and the Greek pollen record? The Chronology section is confusing and focuses on how Marino et al. (2015) have dated ODP site 975. The authors only provide in Table S3 (supplementary information) the common age control points between GDEC-4-2 and ODP 975 for TII and MIS 5e, but they do not refer to the related stratigraphic events. What are the control points for dating TI, TIII and the MIS 1, MIS 5e, MIS 7e, and MIS 7c warming peaks? Based on the issues detailed above, I recommend major revisions before any potential publication.

Other comments Page 2, line 25 – But Toucanne et al. (2015) suggested that the enhanced rainfall in the western Mediterranean during warm periods of the last interglacial was regional and due to the intensification of winter Mediterranean storm tracks.

Page 9, line 14 - The authors describe the paleoclimatic data of MIS 6/5 and MIS 2/1 transitions related to HS11 and HS1, respectively, but not those of MIS 8/7 and MIS 7d/7c that, although not related to Heinrich events, are associated with IRD pulses as shown by ODP 980 (McManus et al., 1999, Science). In Figure 3 (to be renamed

Figure 2) the authors should add the intervals of IRD deposits.

Page 3, line 5; Page 4, lines 25-26 – Why do the authors cite Raislback et al., 2015 for MIS 5 and MIS 5e and not for MIS 7c and MIS 7e?

Page 3, lines 5-6 – Fletcher and Sánchez Goñi (2008) article presents a marine pollen sequence. Therefore, it is not a lacustrine record as indicated by Dixit et al.

Page 4, lines 1-11 – The authors should improve Figure 1 by representing the hydrography affecting GDEC-4-2 site.

Page 5, line 11 – Add a "," after "sporadically" and "to" before "obtain".

Page 5, line 26; legend of Figure 3 – Replace "precession" with "precision".

Page 6, lines 22-23 – Delete this sentence. It is a repetition of lines 19-21.

Page 6, lines 23-25 – Water temperatures can be also decoupled from local atmospheric temperatures during periods of ice-growth (Sanchez Goñi et al., 2013, Nat. Geosci.), not only during periods of high river discharges.

Page 7, line 12 – Modify the following sentence "…a reduced concentration of atmospheric concentrations…".

Page 7, line 19 – Replace "isotopes" with "record".

Page 8 – lines 10-12 – Rephrase this sentence, and move all section 3.1 to section 2 "Material and Methods".

Page 9, line 4 – Add "," before "respectively".

Page 9, lines 5-6 – Delete "Ocean Drilling Program" and "s" from "Sites".

Page 9, line 12 – Replace "ice-sheet" with "iceberg".

Page 9, line 22 – Figure S4 is not necessary. The same information is presented in Figure 3.

Page 10, lines 13-14 – Contrary to authors' statement, d18Osw values are not shown for site LC21 in Figure 2g. Only d18O values of G. ruber are represented.

Page 10, line 13 – Delete "s" from "sites".

Page 10, lines 14-16 – Add "through the Nile river".

Page 10, line 18 – Delete "at".

Page 10, lines 21-22 – Delete "and sea level changes".

Page 10, lines 22-26 – Please rephrase this sentence.

Page 11, lines 3 and 27 – Replace "Fig. 3f" with "Fig. 3e".

Page 12, line 2 - Replace "Fig. 3f" with "Fig. 3e".

Page 12, line 16 – Add "during winter" after "North Atlantic region".

Page 12, line 24 – Amore et al, 2012 is not listed in the references.

Pge 12, line 27 – The "increased precipitation" of the mid-Holocene occurred in winter?

Page 14, line 3 – Delete "direct". The geochemical evidence presented by the authors is not a direct evidence for winter precipitation.

Page 15, line 4 – Replace "analysis" with "analyses".

Page 26, line 3 – Replace "...for the last MIS 5e" with "for TII and MIS 5e".

Page 26, lines 7-8 – From where do pollen records come?

Page 26, lines 9-10 – Vertical orange and yellow bars do not indicate "warmer substages of the last interglacial". Both bars are within the Marine Isotopic Substage 5e.

Page 27, lines 7-8 – Contrary to the authors' statement, vertical orange and yellow bars do not indicate stadials during MIS 1 and MIS 5. Also there is no orange bar. Please harmonize the colors between figures 2 and 3.

In Figure 2, the blue band indicating the HS1 should be enlarged and start at 18 ka and not at 17 ka as shown.

---

## Author Comment (AC2) · 14 Nov 2019

We cordially thank the reviewer for their constructive comments and helpful suggestions on the manuscript. Our replies to each of the comments are given in blue.

**Major comments**

The manuscript submitted by Dixit et al to the Climate of the Past journal is a revised version of a manuscript that I reviewed for another journal almost two years ago, and I must say that several of my concerns have not been suitably addressed. Dixit et al. present Mg/Ca and Ba/Ca-based reconstructions of sea surface temperature (SST) and salinity, respectively, for the Tyrrhenian Sea during the last three terminations (TI, TII en TIII) and peak interglacials of MIS 1, MIS 5e, MIS 7e and MIS 7c. From these reconstructions the authors infer changes in Golo River runoff that they interpret to indicate changes in winter rainfall over the northern Mediterranean basin. They observe that the long-term amplitude of the salinity decrease tightly follows eccentricity. They also find that during SST warming putative increases in winter rainfall coincide with increases of African monsoon, increase of Nile River runoff in summer, both developing well-stratified column waters, periods of anoxia and sapropel layers. A comparison of these results with model simulations for the mid-Holocene allows the authors to support the idea of an increased southwesterly moisture transport into the western Mediterranean from the North Atlantic. The first observation is new, and it is interesting to know the factor that may modulate the amplitude of long-term changes in salinity in the Mediterranean Sea; the second finding is not new but supports previous studies explaining sapropel formation (Toucanne et al., 2015, QSR; Grant et al., 2016, QSR) and the origin of the western Mediterranean rainfall during the mid-Holocene (Brayshaw et al., 2011, The Holocene). Therefore, I do not see the real contribution of this manuscript. Moreover, interpreting changes in runoff as direct evidence of changes in seasonal (winter) rainfall seems to me to be inappropriate. Changes in runoff can be the result of changes in vegetation cover with increased runoff during late summer/early autumn in the Mediterranean associated with more erosion, i.e. less forest cover (Durán Zuazo and Rodríguez Pleguezuelo, 2008, Agronomy for Sustainable Development). Additionally, changes in salinity can also result

from changes in precipitation-evaporation balance. These issues are not sufficiently discussed in the manuscript.

Reply: We thank the reviewer for acknowledging the novelty of the results inferred from Ba/Ca- runoff record, that the amplitude of the Mediterranean winter precipitation was controlled by eccentricity-modulated precession and that these results provide further support to previous studies explaining the role of Mediterranean winter rain in sapropel formation. Therefore, now we explicitly state in the manuscript both of these points. Lines 1-21 Page 17 now reads

 "In this study, we used geochemical proxies to better assess the variation in winter rainfall in the western Mediterranean during the Holocene and the past two interglacials. Our geochemical data suggest increased runoff/ rainfall during the warm periods of the Holocene and the past two interglacials.  Proxy data demonstrate that the intensity of the precession-controlled wintertime rainfall in the western Mediterranean was modulated by eccentricity, with times of high eccentricity characterized by higher rainfall and river outflow. These results along with the analysis of Holocene climate simulations support increased winter precipitation sourced from the North-Atlantic in a warmer western Mediterranean during the past. Our data and model results also show that high rainfall events in the northern Mediterranean borderland occurred at times of intensified North African summer monsoon and the sapropel deposition in the Mediterranean basin. This is in agreement with recently published proxy reconstructions for past 1.3 Ma and climate simulations from Lake Ohrid in Central Mediterranean (Wagner et al., 2019). The close chronological correspondence of increased river outflow and winter rainfall to organic carbon deposition and sapropel occurrence supports a causal link. We suggest a close coupling between low and high latitude atmospheric-oceanic processes in triggering anoxia in the basin, with a contribution from, both Nile River outflow changes due to variations in African summer monsoon rainfall as well as North Atlantic climatically-controlled winter-rainfall driving outflow changes in the western Mediterranean."

The main objective of this work was to provide direct geochemical constraints to regional rainfall (salinity) and sea surface temperatures, and to test the hypothesis that increased winter rainfall in the western Mediterranean was North Atlantic- sourced, using PMIP3 model simulations. Increased North Atlantic sourced moisture during past interglacials was previously proposed by some (Kutzbach et al., 2014; Toucanne et al., 2015 and references therein), but contended by several authors (Rohling et al., 2016 for detailed review). We have

now stated the aim clearly in the introduction section and also what distinguishes this work from Toucanne et al., 2015. Lines 20-26, Page 3 now reads:

"Mediterranean climates are characterized by strong seasonal contrasts with dry summers and wet and highest precipitation amounts annually during winters from October to March. This winter rain is highly variable in amplitude (Xoplaki et al., 2004). Changes in winter rainfall are critical for regional socioeconomic development for the Mediterranean region, but there still remains a lot of ambiguity on the pattern and mechanism of winter rainfall variability, specifically on Quaternary timescales (IPCC, 2014). Proxy and model studies suggest increased winter rainfall in the western Mediterranean and northern Mediterranean borderland during the Holocene (Carrión, 2002; Fletcher and Sánchez Goñi, 2008; Magny et al., 2011, 2013; Peyron et al., 2011; Zanchetta et al., 2007; Zielhofer et al., 2017) and also during the MIS (Marine Isotope Stage) 5e (Drysdale et al., 2005; Milner et al., 2012; Regattieri et al., 2014) i.e. the warm period of the last interglacial (Railsback et al., 2015). A putative link between high seasonality and increased winter rainfall in the central Mediterranean has also been suggested for the MIS 5e (Milner et al., 2012). This increased winter precipitation in the Mediterranean is attributed to higher air–sea temperature difference and locally induced convective precipitation that dominate changes in the freshwater budget on obliquity timescales (Bosmans et al., 2015; Rohling et al., 2015; references therein). Alternatively, recent study from the oldest lake in Europe, Lake Ohrid in the Balkan Peninsula show that high North Atlantic sourced moisture into the Mediterranean during winters was the primary driver of Mediterranean hydrological changes both on precessional and seasonal timescales during the interglacials of the past 1.36 Ma (Wagner et al., 2019). The Atlantic signature in Mediterranean precipitation is also visible during MIS 10-11 in a leaf-wax isotopic record from Tenaghi Philippon peatland, NE Greece (Ardenghi et al., 2019). In this light, direct rainfall/sea surface salinity (SSS) and sea surface temperature (SST) estimates from well-located, regionally representative archive that cover multiple interglacial periods, is key to addressing long-standing questions regarding the underlying mechanisms and amplitude of winter precipitation variability.

In this study, we bridge the gap by investigating SSS and SST changes in the marine sediment core GDEC-4-2 located off eastern Corsica (Fig. 1). Previously, Toucanne et al. (2015) used sediment characteristics from GDEC-4-2 to propose enhanced North Atlantic-sourced rainfall in the Western Mediterranean during warm intervals of interglacial periods over the last 547 ka BP. Here, we develop independent geochemical record to assess

precipitation variability by reconstructing runoff (rainfall/salinity) and temperature changes at the GDEC-4-2 site during the Holocene (MIS 1), the last (MIS 5) and penultimate (MIS 7) interglacials using trace element and stable isotopes of the planktonic foraminifera *Globigerina bulloides* (Fig. 1). We then compare our geochemical proxy records with PMIP model simulations and prominent Mediterranean records, to provide a mechanistic understanding of interglacial precipitation variability."

We agree with the reviewer that changes in runoff could indeed have been affected by seasonal changes in local vegetation, however at longer timescale, sub-orbital in this case, this seasonal effect would be masked by the enhanced precipitation signal which will be reflected in Ba/Ca data. We do concur that the salinity changes can be a result of local evaporation/precipitation, which is the reason why the use of $\delta_{18}O$ is limited in such cases as it is unable to distinguish between the local evaporation/precipitation signal from the global ice-volume and ensuing sea level changes. Therefore, we used Ba/Ca in foraminifera calcite as an independent proxy for local runoff changes. We now discuss this in details on Page 10, lines 20-25 which read
"Difficulties in extracting the local evaporation/precipitation signal from the global ice-volume and ensuing sea level changes, limits the use of $\delta_{18}Osw$ to examine local hydrological changes. Therefore, at GDEC-4-2 site, the $\delta_{18}Osw$ reflect a combined signature of global ice-volume changes and changes in regional precipitation and the Ba/Ca record for river discharge provide independent estimates of the change in SSS for the past interglacials, with high Golo runoff implying increased local precipitation during warm intervals in the western Mediterranean."

Based on previous research in the Mediterranean region, they state that other proxies (pollen and speleothems) are "unable to offer direct insights on the variability in winter rainfall". Contrary to this statement and as far as pollen studies are concerned, we know that present-day changes in Mediterranean forest cover depend on the North Atlantic Oscillation shifts, i.e. on the position and intensity of the westerlies that in turn control winter precipitations in Europe (Gouveia et al., 2008, Int. J. of Climatology). Therefore, pollen-based Mediterranean forest cover changes are direct evidence of changes in winter precipitation as repeatedly shown by data (Fletcher and Sanchez Goñi, 2008, Quat. Res.) and model–data comparisons for different interglacials of the last 800,000 years (Peyron et al., 2017, Climate of the Past;

Oliveira et al., 2018, Climate Dynamics). Moreover, some of these records have allowed for quantitative reconstructions of winter precipitation for TI and the peak of MIS 1 (Fletcher et al., 2010, Climate of the Past; Peyron et al., 2017). I am surprised by the fact that the authors refer to some of these papers in the Discussion section to support their interpretation after criticizing such an approach in the Introduction. Moreover, they justify their work by the inability of this proxy to reconstruct winter rainfall.

Reply: We agree with the reviewer that pollen studies indeed provide direct insights (for example, Fletcher and Sanchez Goñi, 2008, Milner et al., 2012), and have cited these studies in the discussion. However, direct sea surface salinity (SSS) and sea surface temperature (SST) estimates from the western Mediterranean Sea for the past three interglacial including the Holocene, cannot be obtained by palynological studies and is warranted to understand the winter rainfall variability in the Mediterranean. We understand that it was not clearly stated in the previous version, we have now clearly stated that we bridge this gap by presenting new SSS and SST record using marine sediment core GDEC-4-2.

Following the suggestions by the reviewer, we have now rephrased the introduction largely to clearly state previous work done and the aim of this paper. We now state this in lines 1-26, page 3.

Throughout the manuscript the authors are not consistent when they refer to the region of precipitation. Sometimes they refer to northern Mediterranean rainfall, at other times to western Mediterranean rainfall, and they discuss records coming from the east and to the west of this region. This inconsistency is problematic as several studies show that climate during the Holocene in the Mediterranean region presents west-east and north-south gradients (e.g. Dormoy et al., 2009, Climate of the Past).

Reply: We thank the reviewer for pointing this inconsistency; we now refer the region as western Mediterranean throughout the manuscript to maintain homogeneity. Please see lines 16, Page 5; line 2 Page 6; line 12 Page 11, for example.

We do also take note that there is a west-east and north-south gradients in the Mediterranean region, which is in line with our model simulations and this is now mentioned in the text (Page 13, line 16-20). This reads:

"Interestingly, the east-west and north-south gradient in precipitation pattern as noted by previous studies (for example, Dormoy et al., 2009; Magny et al., 2013) is consistent with the increased south-westerly transport in the region, such that the records showing wetter mid-Holocene lie in the stippled area of our simulation results indicating increased moisture."

I am also concerned by how the authors deal with the timing of Terminations. Terminations are intervals from glacial to interglacial states that generally last a few thousand years and are not events (midpoints) as suggested by Dixit et al. (dashed line in Figures 2 and 3). Terminations should be identified from the 18O of benthic foraminifera, and they are triggered by a combination of ice volume and orbital parameters (Parrenin and Paillard, 2012, Climate of the Past). Cheng et al. (2009, Science) established the timing of marine oxygen isotope terminations (18O of benthic foraminifera) by correlating North Atlantic ice rafted debris (IRD) to radiometrically dated oxygen-isotope cave records from China. The timings of the onset and end of Terminations I, II and III are 18-11 ka, 138-129 ka, 251-243 ka, respectively, and the timing of the midpoint terminations are 14.5 ka, 131 ka, 247 ka. These accurate measurements of the timing of terminations do not coincide with the dates given by Dixit et al. The authors say that the three terminations are centered on 11, 129 and 243 ka, but they do not specify how they established them. With respect to this issue, I invite the authors to look at the recent paper by Barker et al. 2019 in Paleoceanography and Paleoclimatology.

Reply: Based on reviewers' suggestion, we have now changed the timing of Terminations following the recent Barker et al., 2019 paper, please see lines 10-11, Page 9. Following Barker at al., 2019, we have also now marked the mid-points of terminations in red dashed line on the figures.

Overall, the organization of the manuscript and the order of the figures should be changed, and the English improved.

Reply: We have now enlarged all the figures and also introduced a new Figure 2 (see below), which has the results for GDEC-4-2 only. Consequently, old figure 2 is now Figure 3, which shows the comparison of Last interglacial studies and GDEC-4-2 results on a new age scale from Marino et al., (2015). Original Figure 3 is now Figure 4 showing comparison of all the three interglacials with other studies. We have kept the model simulation results (original Figure 4 now Figure 5) as part of our article, as we think this is important to understand the

source of winter rains and complements other model and isotopic studies tracking the source of moisture from the North Atlantic into the Mediterranean, as also discussed above in detail. We have also now checked and corrected the manuscript for any organizational and English grammatical errors. Additionally, we have also modified Figure 1 and its caption and added lake level records used in Figure 4. New Figures 1 and 2 are shown at the end of this document.

For instance, the environmental setting and the studied material are explained twice: at the end of the introduction and at the beginning of the Material and Methods.
Reply: We thank the reviewer for pointing this repetition – this is now corrected.

The introduction should be more focused and clearly explain the gap this work aims to fill and justify the interest of working and comparing the MIS 1, MIS 5e and MIS 7e and MIS 7c warm periods and the related Terminations.
Reply: We have made several changes in the Introduction to clearly state the aims of the paper and the existing gap in our knowledge of hydroclimate variability in the western Mediterranean. Please see previous reply.

Adding model simulations of precipitation changes during contrasting MIS 5e, MIS 7c and MIS 7d interglacials could be relevant.
Reply: We agree with the reviewer that model simulations for MIS 5e, MIS 7c and MIS 7e would have been quite relevant, however, our current understanding of past interglacials is limited because previous coordinated model intercomparisons do not include interglacials older than the Holocene, though this is a topic of active work with PMIP4 simulations will cover the last interglacials available in the coming years (Otto-Bliesner et al., 2016). The idea was, however, recently tested for the interglacials in the past 1.3Ma, from Lake Ohrid on the Balkan Peninsula, together with transient climate model simulations and proxy time series, and they confirm that increased sea surface temperatures during the interglacial amplify local cyclone development and refuel North Atlantic low-pressure systems that enter the Mediterranean when continental ice volume was low and concentrations of atmospheric greenhouse gases were high (Wagner et al., 2019). Therefore, we have added these details to the discussion 1-9, Page 13, with this caveat and pointing out that this will be testable with future simulations. We have now also modified Figure 4 and added recently published pollen data and percent Total inorganic carbon (TIC%) from Lake Ohrid (Wagner et al., 2019,

Nature), which indicate higher aquatic productivity due to warmer conditions and pollen analyses show a simultaneous increase in vegetation cover during early phases of the Last and the Penultimate interglacial periods. This is consistent with our Ba/Ca- river runoff record and PMIP3 model simulations analyzed in our study.

The subsection "Proxy systematics" should be moved to the Material and Methods section. Furthermore, I do not understand the meaning of "systematics" in this context.
Reply: We have made this modification and have moved the 'Proxy systematics' subsection to Materials and Methods. The term 'Proxy systematics' is used to explain the mechanism of how the temperature and rainfall proxies work.

Figure 3 in which all the results from this study are shown should be Figure 2. Figure 2 is only displaying different records covering TII and MIS 5e in the Mediterranean.
Reply: We have now enlarged all the figures and also introduced a new Figure 2, which has the results for GDEC-4-2 only. Consequently, old figure 2 is now Figure 3, which shows the comparison of Last interglacial studies and GDEC-4-2 results on a new age scale from Marino et al., (2015). Original Figure 3 is now Figure 4 showing comparison of all the three interglacials with other studies. We have also modified Figure 4 and added recent pollen data and weight% Total Inorganic Carbon data from Lake Ohrid in the Central Mediterranean for the studied intervals (Wagner et al., 2019, Nature). We have kept the model simulation results (original Figure 4 now Figure 5) as part of our article, as we think this is important to understand the source of winter rains and complements other model and isotopic studies tracking the source of moisture from the North Atlantic into the Mediterranean, as also discussed above in detail.

Additionally, I have found many inconsistencies throughout the manuscript, sentences difficult to understand, and several typographic mistakes (see below other comments). In the conclusion section I have one major concern related to the following sentence: "Proxy data placed on a globally synchronous timescale demonstrate that the intensity of the precession-controlled wintertime rainfall: : :": What do the authors mean by "globally synchronous timescale"? How have the authors harmonized the different paleoclimatic records presented in the work: GDEC-4-2, ODP sites 975 and 976, Corchia and Tana Urla speleothems, and the Greek pollen record? The Chronology section is confusing and focuses on how Marino et al. (2015) have dated ODP site 975. The

authors only provide in Table S3 (supplementary information) the common age control points between GDEC-4-2 and ODP 975 for TII and MIS 5e, but they do not refer to the related stratigraphic events. What are the control points for dating TI, TIII and the MIS 1, MIS 5e, MIS 7e, and MIS 7c warming peaks?

Reply: We have now corrected the inconsistencies and typos as pointed by the reviewer. For the chronology, we used the age model constructed by Toucanne et al., 2015 for the Holocene and MIS 7c and 7e, which can be referred for more details regarding tie-points, glacial terminations and warm intervals of MIS 1, MIS 7e and MIS 7c.

For MIS 5e, however, we used the chronology constrained by Marino et al., (2017) as they present a new radiometrically constrained chronology for North Atlantic records of climate variability which they synchronized with ODP sites 976, Corchia and Tana Urla speleothems and also with the marine records from the eastern Mediterranean exploiting the well documented intermediate-water connectivity between the eastern and western Mediterranean Sea. Since we synchronized our MIS 5e chronology with ODP975 chronology, our chronology is in turn synchronized with the records in Figure 3. To clarify, we now state in the text

"We followed the chronology described in Toucanne et al. (2015), which is constrained by aligning the planktonic δ18O, weight percent CaCO3 and XRF-Ca/Ti to the NGRIP ice core $\delta_{18}$O record from Greenland for the last glacial termination (GICC05 chronology; Rasmussen et al., 2006; Svensson et al., 2008) and to the synthetic Greenland (GLTsyn) record of Barker et al. (2011) from 60 to ~550 ka BP. For penultimate glacial termination T-II and MIS 5, we synchronized $\delta_{18}$O of G. bulloides to the most up-to-date radiometrically-constrained chronology of ODP Site 975 (Marino et al., 2015) (Fig. 3a; Table S2). Marino et al. (2015) obtained a new radiometrically constrained chronology for ODP975 across T-II and the last interglacial period exploiting the well-documented intermediate-water connectivity between the eastern and western Mediterranean Sea, and the relationship between marine surface water microfossil $\delta_{18}$O and U-series-dated regional $\delta_{18}$O speleothem records. This was done to obtain a regionally (both eastern and western Mediterranean) synchronous picture for this time period. The $\delta_{18}$O of planktonic foraminifera G. bulloides from the site ODP 975 is synchronized to the Soreq Cave speleothem and $\delta_{18}$O of G. bulloides from marine core LC21 in the eastern Mediterranean, and to $\delta_{18}$O of G. bulloides of ODP Sites 976, 977 and core MD01-2444 in the western Mediterranean, thereby to the SST and/or IRD records of North

Atlantic climate variability that are archived in the Iberian margin sediment cores (see supplementary information, Table S2).

Other comments

Page 2, line 25 – But Toucanne et al. (2015) suggested that the enhanced rainfall in the western Mediterranean during warm periods of the last interglacial was regional and due to the intensification of winter Mediterranean storm tracks.

Reply: We agree with the reviewer in that Toucanne et al., (2015) did hypothesize that the winter rainfall in the western Mediterranean was due to intensified storm tracks, while Rohling et al., (2015) ruled out any extra Mediterranean moisture source and suggested the winter rainfall is recycled moisture from the basin itself. This is in fact the primary objective of this study to provide quantitative SST and SSS estimates such Toucanne et al. hypothesis can be investigated. We do realize that it was not very clear in our earlier version, hence we have now revised our paper to state this point clearly.

Lines 20-26, Page 3 now read

"In this study, we bridge the gap by investigating SSS and SST changes in the marine sediment core GDEC-4-2 located off eastern Corsican (Fig. 1). Previously, Toucanne et al. (2015) used sediment characteristics from GDEC-4-2 to propose enhanced North Atlantic-sourced rainfall in the Western Mediterranean during warm intervals of interglacial periods over the last 547 ka BP. Here, we develop independent geochemical record to assess precipitation variability by reconstructing runoff (rainfall/salinity) and temperature changes in the western Mediterranean basin during the Holocene (MIS 1), the last (MIS 5) and penultimate (MIS 7) interglacials using trace element and stable isotopes of the planktonic foraminifera Globigerina bulloides (Fig. 1). We then compare our geochemical proxy records with PMIP model simulations and prominent Mediterranean records, to provide a mechanistic understanding of precipitation variability."

Page 3, line 5; Page 4, lines 25-26 – Why do the authors cite Raislback et al., 2015 for MIS 5 and MIS 5e and not for MIS 7c and MIS 7e?

Reply: We have now made corrections in the citations which now reads. "Here, we develop independent geochemical record to assess precipitation variability by reconstructing runoff (rainfall/salinity) and temperature changes in the western Mediterranean basin during the Holocene (MIS 1), the last (MIS 5) and penultimate (MIS 7) interglacials (Railsback et al.,

2015) using trace element and stable isotopes of the planktonic foraminifera Globigerina bulloides (Fig. 1).

Page 3, lines 5-6 – Fletcher and Sánchez Goñi (2008) article presents a marine pollen sequence. Therefore, it is not a lacustrine record as indicated by Dixit et al.

Reply: We have corrected this point now (Page 1, line 23-25) which reads. "Proxy and model studies suggest increased winter rainfall in the Mediterranean during the Holocene (Carrión, 2002; Fletcher and Sánchez Goñi, 2008; Magny et al., 2011, 2013; Peyron et al., 2011; Zanchetta et al., 2007; Zielhofer et al., 2017) and also during the MIS (Marine Isotope Stage) 5e (Drysdale et al., 2005; Milner et al., 2012; Regattieri et al., 2014) i.e. the warm periods of the last interglacial (Railsback et al., 2015)."

Page 4, lines 1-11 – The authors should improve Figure 1 by representing the hydrography affecting GDEC-4-2 site.

Reply: We have now added the text describing the hydrography affecting the site GDEC-4-2. We think that the figure would become very crowded if we add the hydrography along with all the existing records and show the connection with the North Atlantic. Since the hydrography is not directly relevant to the scope of this work, we have now added references (Toucanne et al., 2012, 2015) for detailed information on the hydrography affecting GDEC4-2 and the region in general.

The text now reads "The hydrography at GDEC-4-2 site is mainly influenced by the Levantine Intermediate Water (LIW) circulation (from ca. 200 to 600-1000 m water depth) ( see Toucanne et al., 2012 for detail information on hydrography of this region). The LIW is formed in the Levantine Basin (eastern Mediterranean) in a permanent large-scale cyclonic Rhodes gyre through summer evaporation and winter cooling (i.e. buoyancy loss; Lascaratos et al., 1999; Malanotte-Rizzoli et al., 2003; Robinson et al., 1992). It forms the major water mass flowing from the east to west, along with the Aegean and Adriatic water contributions. In the northern Tyrrhenian Sea, a portion of the LIW flows northwards through the Corsica Trough, while the other part flows southwards to the Sardinia Channel, then along the western slope of Sardinia and Corsica before its intrusion into Ligurian Sea. The LIW contributes to the Western Mediterranean Deep Water production after reaching the Gulf of Lion and both water masses contribute to ca. 80% and 20% to the Mediterranean Outflow water (MOW), respectively (Pinardi and Masetti, 2000).

Page 5, line 11 – Add a "," after "sporadically" and "to" before "obtain".

Page 5, line 26; legend of Figure 3 – Replace "precession" with "precision".

Reply: Both the errors are corrected now.

Page 6, lines 22-23 – Delete this sentence. It is a repetition of lines 19-21.

Reply: We have now taken out this sentence to avoid repetition.

Page 6, lines 23-25 – Water temperatures can be also decoupled from local atmospheric temperatures during periods of ice-growth (Sanchez Goñi et al., 2013, Nat. Geosci.), not only during periods of high river discharges.

Reply: We agree with the reviewer that a decoupling between the atmosphere and water temperature does occur during periods of ice-growth, however the decoupling observed in our record occurs during globally warm period and hence we suggest that the decoupling is an artifact of high river discharges. This now reads

"These globally warm periods are characterized by increased Mg/Ca-based-SSTs at GDEC-4-2 with ~18 ℃ Holocene values, and MIS 5e being the warmest with temperatures averaging ~24℃, although at this site, our SST reconstructions indicate increased riverine discharge during MIS 7c and 7e led to local SST being cooler (Fig. S4)."

Page 7, line 12 – Modify the following sentence ": : :a reduced concentration of atmospheric concentrations: : :".

Reply: This sentence is now modified to read "Mid-Holocene conditions differ from the PI period through their orbital configuration and reduced atmospheric greenhouse gas concentrations" on lines 24-25, Page 7

Page 7, line 19 – Replace "isotopes" with "record".

Reply: We have corrected this now.

Page 8 – lines 10-12 – Rephrase this sentence, and move all section 3.1 to section 2

Reply: We have made the changes suggested by the reviewer. The sections now read

"2.5 Proxy systematics

The $\delta_{18}O$ in foraminiferal calcite is controlled by calcification temperature and the $\delta_{18}Osw$ (Bemis et al., 1998). $\delta_{18}O$ sw was estimated using Mg/Ca-based-SST in concert with analysed calcite δ18O ($\delta_{18}O$ calcite) for *G. bulloides* ($\delta_{18}O$ *G. bulloides*) and temperature - (δ18O calcite - $\delta_{18}Osw$) relationship (Bemis et al., 1998). $\delta_{18}Osw$ in turn is controlled by

salinity variations due to river runoff, and changes in the isotopic composition of river water and seawater. The latter in turn reflects changes in precipitation relative to evaporation, advection of surface waters to the site, and continental ice volume changes. Foraminiferal Ba/Ca is used as an independent proxy for riverine runoff and rainfall changes (Weldeab et al., 2007). Seawater Ba (Basw) concentrations at sites influenced by riverine discharge are highly correlated to salinity because dissolved Ba is high in riverine water and Ba desorbs from suspended sediments in estuaries (Coffey et al., 1997). Ba incorporation in foraminiferal calcite varies linearly with changes in Basw and is therefore independent of temperature and alkalinity (Hönisch et al., 2011). We also attempted to use the modern Ba/Casw-salinity relationship obtained off the Golo River to obtain a first-order estimate of the past runoff-induced SSS variations, as recorded by Ba/Caforam."

"Material and Methods".
Page 9, line 4 – Add "," before "respectively".
Page 9, lines 5-6 – Delete "Ocean Drilling Program" and "s" from "Sites".
Page 9, line 12 – Replace "ice-sheet" with "iceberg".
Page 9, line 22 – Figure S4 is not necessary. The same information is presented in Figure 3.
Reply: We have made all the above changes suggested by the reviewer.

Page 10, lines 13-14 – Contrary to authors' statement, d18Osw values are not shown for site LC21 in Figure 2g. Only d18O values of G. ruber are represented.
Reply: The reviewer is correct, we indeed mean $\delta^{18}O$ of foraminifera and not seawater, which is shown in figure 2g. We have corrected this error now.

Page 10, line 13 – Delete "s" from "sites".
Reply: This has been corrected now.
Page 10, lines 14-16 – Add "through the Nile river".
Reply: We have added "through the Nile River and other north African rivers"

Page 10, line 18 – Delete "at".
Page 10, lines 21-22 – Delete "and sea level changes".

Reply: We have corrected these two points now.

Page 11, lines 3 and 27 – Replace "Fig. 3f" with "Fig. 3e".
Page 12, line 2 - Replace "Fig. 3f" with "Fig. 3e".
Reply: We have corrected these two points now.

Page 12, line 16 – Add "during winter" after "North Atlantic region".
Reply: We have corrected this now.

Page 12, line 24 – Amore et al, 2012 is not listed in the references.
Reply: We have added Amore et al., 2012 reference.

Pge 12, line 27 – The "increased precipitation" of the mid-Holocene occurred in winter?
Reply: We agree with the author, it is indeed "increased winter precipitation during the mid-Holocene". We have corrected this now.

Page 14, line 3 – Delete "direct". The geochemical evidence presented by the authors is not a direct evidence for winter precipitation.
Reply: We agree with the author and have corrected this now.

Page 15, line 4 – Replace "analysis" with "analyses".
Reply: We have corrected this now.

Page 26, line 3 – Replace ": : :for the last MIS 5e" with "for TII and MIS 5e".
Reply: We have corrected this now

Page 26, lines 7-8 – From where do pollen records come?
Reply: The pollen records come from Tenaghi Philippon peatland in NE Greece. We have added this detail in the figure caption now.
Page 26, lines 9-10 – Vertical orange and yellow bars do not indicate "warmer substages of the last interglacial". Both bars are within the Marine Isotopic Substage 5e.
Reply: We have reworded the sentence to read "Vertical light yellow bars indicate interglacial conditions *s.l.* and dark yellow bars denote interglacials warm intervals and the interglacial *s.s.*"

Page 27, lines 7-8 – Contrary to the authors' statement, vertical orange and yellow bars do not indicate stadials during MIS 1 and MIS 5. Also there is no orange bar. Please harmonize the colors between figures 2 and 3.

Reply: We have made these corrections in the figures now.

In Figure 2, the blue band indicating the HS1 should be enlarged and start at 18 ka and not at 17 ka as shown.

Reply: We have now corrected HS1 in Figure 3.

**Modified Figure 1 and new figure 2:**

[Figure]

Figure 1: Location of GDEC-4-2 (red) in the Northern Tyrrhenian Sea and other marine (blue) and terrestrial archives (green). Numbers and black dots denote the lake level records used to compare results with model simulations in Figure 5. 1) Lake Medina in southern Spain (Reed et al., 2001); (2) Lake Siles in southern Spain (Carrión, 2002); (3) Lake Cerin (Magny et al., 2011); (4) Lake Ledro in northern Italy (Magny et al., 2012); (5) Lake Accesa in central Italy (Magny et al., 2007); (6) Lake Grande diMonticchio in Basilicata, southern Italy (Allen et al., 1999); (7) Lake Albano and Lake Nema (Ariztegui et al., 2000); (8) Lake Preola in Sicily (Magny et al., 2011); (9) Lake Xinias in northern Greece (Digerfeldt et al., 2007); (10) Lake Golhisar in south-western Turkey (Eastwood et al., 2007); (11) Lake Eski Acigol in central Turkey (Turner et al., 2008); (12) Lake Van in Turkey (Pickarski and Litt, 2017). Red band and red dotted line denotes the extent of modern summer ITCZ and the maximum northward reach of ITCZ in the past respectively (Tuenter et al., 2003). Also shown are the sea-level pressures in North Atlantic and the direction of Mediterranean storm tracks (black).

[Figure]

**Figure 2: GDEC-4-2 results for the last three interglacials. (a)** δ18O *G. bulloides*; **(b) Mg/Ca-based SSTs from *G. bulloides* (blue); (c) Ba/Ca in foraminifera as a proxy of river discharge. Vertical light yellow bars indicate interglacial conditions *s.l.* and dark yellow bars denote interglacials warm intervals and the interglacial *s.s.* Sapropel deposition intervals, Heinrich stadials (blue bar) and mid-points of glacial terminations (dashed red line, following Barker et al., 2019) shown on top.**

---

## Author Comment (AC1)

We cordially thank the reviewer for their constructive comments and helpful suggestions on
the manuscript. Our replies to each of the comments are given in blue.

The paper from Dixit et al present trace elements (Ba/Ca and Mg/Ca) and stable
oxygen isotope composition from planktic foraminifera (G. bulloides) from the previously
published marine core GDEC-4-2 from the Corsica margin. The data cover the
Holocene, the interglacials MIS5 and MIS7 and the glacial termination TI, TII, TIII. Sea
surface temperature were obtained from Mg/Ca data and used together with _18O calcite
data to calculate _18O of sea water. Ba/Ca is used as a proxy for Golo River
discharge and, through calibration using the modern sea surface salinity (SSS)-Ba/Ca
relationship, the authors attempt to quantitatively reconstruct past SSS. Using these
data, and by comparison with several records (both marine and continental) from the
Mediterranean, the authors suggest that the three interglacial were characterized by an
increase in winter precipitation driven by changes in North Atlantic storm tracks trajectories,
in turn modulated by changes in precession and eccentricity. To support their
hypothesis, the authors also analysed outputs from modelling experiments (PMIP3) for
the pre-industrial Holocene. Authors also suggests that these increases in precipitation
contributed to trigger basin anoxia and sapropel deposition. The presented data are of
interest and their interpretation as paleo-rainfall variability proxies is reasonable. However,
main text, figures and supplementary material are rather confused and misleading
in some points, references are not updated and often messed-up, the comparison with
the model almost useless and the whole discussion is a bit inconsistent and not fully
supported by the data. Moreover, the main findings of the paper add very few to what
was already proposed in the original paper on the same core (Toucanne et al., 2015).
Thus, I suggest publication in Climate of the Past only after careful major revision.

Reply: We have now significantly restructured the manuscript following reviewers'
comments. Our replies to each of reviewer's point are given in blue.

Main Points:

One of the main claims of the paper is that variations in winter precipitation

are modulated by eccentricity changes. I found this claim a bit obvious. Indeed,

it is well known from both data and modelling experiments that winter precipitation in

the Mediterranean are mostly modulated by precession changes (e.g. Tzedakis et al.,

2007; Milner et al., 2012; Toucanne et al., 2015; Regattieri et al., 2015; Bosmans et

al., 2015, just to quote some, but there are others). The intensity of the precession

forcing relate to changes in eccentricity, with higher eccentricity inducing higher precession

forcing and lower eccentricity reducing the influence of this orbital parameter.

Also, the importance of obliquity changes is not mentioned at all, while several works (se e.g.

Bosmans et al., 2015 and references therein) have showed that it has an impact on the

Mediterranean hydroclimate. I think that authors have to largely re-focus the discussion,

better explaining the relationship between eccentricity and precession and taking into account

the influence of obliquity changes. To this end, I suggest that they have a detailed look to

Bosmans et al (2015) results.

Reply: We agree that previous studies mentioned by the reviewer show increased winter
precipitation during past interglacials were forced by precession- derived boreal insolation
forcing. These studies are cited in the various parts of the manuscript (for example, line 21-
25, Page 10; line 10-13, Page 12). Our Ba/Ca record from western Mediterranean provides
direct geochemical evidence for increased precessionally-paced rainfall tightly linked with
eccentricity cycle during the last three interglacials from the same site. We do agree that
increased insolation seasonality, such as minimum precession and maximum obliquity has
been previously proposed, for intensification of African summer monsoon and also for
Mediterranean winter precipitation (Bosmans et al., 2015a, b), this information was missing
in our previous discussion. We have now added this information with relevant references in
line 9-12, Page 3, which now reads

"This increased winter precipitation in the Mediterranean is attributed to higher air–sea
temperature difference and locally induced convective precipitation that dominate changes in
the freshwater budget on obliquity timescales (Bosmans et al., 2015; Rohling et al., 2015;
references therein)."

As suggested by the reviewer, we have also added this in the discussion (line 19-22, Page
12).

Another very weak point is the claim that the findings of the paper are supported by

modelling results. I found this part confused and even misleading. First because the model output is related only to the mid-Holocene, so results cannot be extended to others interglacials where the boundary conditions were so different, second because the whole discussion about changes in storm tracks trajectories and NAO like atmospheric patterns are not supported, to me, neither by the data or by the modelling experiment that they present. I can agree that the Mid-Holocene experiments shows a southern shift in westerly trajectories which can resemble a NAO- pattern, but I do not see any reason to extend this interpretation to the other considered periods. The most likely mechanism for increased precipitation during precession minima, basing on available data and literature, is related to changes in Mediterranean-sourced precipitation due to increased Med heat content. Indeed, during precession minima hydrology in the Med is influenced by low-latitudes atmospheric patterns: the northward shift of the ITCZ causes stronger summer drought related to the descendent branch of the Hadley cell. It causes an increase in the Mediterranean summer heat content. High summer temperatures lead to elevated sea-surface temperatures and associated high evaporation levels persisting well into the year, contributing to the formation of depressions across the northern borderlands, strengthening cyclogenesis within the basin and causing an increase in autumn-winter precipitation. This is what has been proposed also basing on GDEC data by Toucanne et al just few years ago, and I do not understand why authors now invoke a completely different mechanism: : :. The whole discussion about the model experiment is very confused, do not add nothing to the interpretation and do not support what the paper claims. I suggest to largely modify section 4.2 trying to explain the mechanisms more relying on the presented data and on previous literature. They should briefly review mechanisms proposed by e.g. Tzedakis et al. (2007) or by Milner et al. (2012) or by Bosmans et al. (2015) and especially by Toucanne et al., 2015, trying to better highlight which one best fits with their results. To me, this whole part about modelling is an, almost failed, attempt to add something new to the -good- explanations already proposed by Toucanne et al. Authors should be more "honest" with that in the sense that they should clearly state that this work is an update of the previous one and that the new data and results strengthen the previous interpretation, without striving to introduce new and confused mechanisms for that.

Reply: Our current understanding of past interglacials is limited because previous coordinated model intercomparisons do not include interglacials older than the Holocene, though this is a topic of active work with PMIP4 simulations will cover the last interglacials

available in the coming years (Otto-Bliesner et al., 2016). We do propose that a similar mechanism was probably responsible for increased winter rainfall during the last interglacial and the penultimate interglacial. This is supported by a recent proxy data and transient climate model study from Lake Ohrid, which show that during the past interglacials of the last 1.3Ma, increased North Atlantic low-pressure systems entered and brought winter precipitation in the Mediterranean when continental ice volume was low and concentrations of atmospheric greenhouse gases were high (Wagner et al., 2019).

However as suggested by the reviewer, we have modified this section largely, with detailed discussion on previously published studies along with recent publications from central and northern Mediterranean and also highlighting specifically our contribution to the debate. The modified section read as follows:

"We used climate model simulations from the Paleoclimate Model Intercomparison Project – Phase 3 (PMIP3)(Braconnot et al., 2011) to shed light on the variability of winter precipitation during these times and also to examine the source of the wintertime Mediterranean rainfall. Our model analysis for a representative interglacial (the mid-Holocene at ~6 ka (as used in PMIP3) compared to pre-industrial (PI) conditions) suggests enhanced southwesterly mean moisture transport from the North Atlantic causing higher moisture convergence during winters in the Mediterranean, potentially brought about by a south-eastward shift of storm tracks (Fig. 5a) during interglacials, in a negative North Atlantic Oscillation (NAO)-type pattern. This North Atlantic moisture signal in winter precipitation is also observed in lipid isotope record from Tenaghi Philippon in NE Greece for the MIS 10-11 period (Ardenghi et al., 2019). They inferred a constant Atlantic source for the bulk of the moisture arriving to the Mediterranean northern borderlands. Prolonged waning of MIS12 ice sheets is proposed to have maintained colder, fresher surface waters in the North Atlantic which created a sharp meridional SST gradient in a negative NAO like conditions strengthening the storm track (Ardenghi et al., 2019). On the same lines, a similar mechanism was probably responsible for increased winter rainfall during the last interglacial and the penultimate interglacial. Our current understanding of past interglacials is however limited because previous coordinated model intercomparisons do not include interglacials older than the Holocene, though this is a topic of active work with PMIP4 simulations will cover the last interglacials and these will be available in the coming years (Otto-Bliesner et al., 2016). The idea has however recently been tested for the interglacials in the past 1.3Ma, from Lake Ohrid on the Balkan Peninsula and together with transient climate model simulations and proxy time series, it is proposed that during the past interglacials increased

North Atlantic low-pressure systems entered and brought winter precipitation in the Mediterranean when continental ice volume was low and concentrations of atmospheric greenhouse gases were high (Wagner et al., 2019). Our Ba/Ca- river runoff/salinity record and PMIP3 model simulations provides further constraints and greater confidence to previous findings on the mechanistic understanding of the Mediterranean winter rains.

Recent extreme rainfall events over the northern Mediterranean borderlands have a distinct North Atlantic origin of moisture (Celle-Jeanton et al., 2001). Today, NAO is the dominant atmospheric phenomena in the North Atlantic and Mediterranean region during winters (Olsen et al., 2012; Hurrell, 1995; Trigo et al., 2002) (Fig. 1), such that during the negative phase of NAO, storm tracks are shifted southwards that bring wet and mild winters over the southern Europe. Fluctuations in NAO strongly affects the intensity of zonal flows over the North Atlantic (i.e. westerlies), the position of storm tracks and subsequent precipitation amount across Europe and the Mediterranean basin (López-Moreno et al., 2011). Coupled atmosphere–ocean general circulation model suggest that these NAO-type mode of climate variability could also have operated at orbital timescales such as MIS 5e (Lohmann, 2017). Wagner et al., 2019 compared annual cycle of simulated Lake Ohrid precipitation data with modern reanalysis data to show that current drivers of the amount of rainfall in the Mediterranean share similarities to those that drove the reconstructed increases in precipitation in the past i.e. a North Atlantic control on the Mediterranean winter precipitation (Wagner et al., 2019). A North Atlantic connection of winter rainfall on the northern Mediterranean borderland was also suggested previously using palynological proxies from the Iberian margin (Amore et al., 2012) and more recently using geochemical proxies from the Gulf of Lion (Pasquier et al., 2019).

Furthermore, all the mid-Holocene model outputs from our model analysis are in good agreement with the mid-Holocene high lake levels, which indicate increased precipitation minus evaporation (P – E) due to increased winter precipitation during the early-mid Holocene (10-6 ka BP) on the western and northern Mediterranean borderlands (Magny et al., 2013) (Fig. 5b). Interestingly, the east-west and north-south gradient in precipitation pattern as noted by previous studies (for example, Dormoy et al., 2009; Magny et al., 2013) is consistent with the increased south-westerly transport in the region, such that the records showing wetter mid-Holocene lie in the stippled area of our simulation results indicating increased moisture. A similar pattern of wetter winter with a strong seasonal cycle of surface air temperatures during the early Holocene was also observed in previous general circulation model simulations (Brayshaw et al., 2011). In particular, a stronger southwesterly flow

during the winter 6kaBP experiment (compared with the PI control run) was clearly shown such that the northern coast and western Mediterranean received strong precipitation (Brayshaw et al., 2011). Comparison of Holocene proxy-models using regional scale downscaling of a set of global climate model simulations for the Mediterranean region also give consistent results (Peyron et al., 2017).

There is also evidence for stronger seasonality in winter precipitation and P – E during interglacials in the PMIP3 simulations (Fig. S7), due to an intensifying moisture convergence in late winter, as previously suggested by palynological records from Greece and Turkey (Milner et al., 2012; Tzedakis, 2007). Previous modelling experiments demonstrate increased winter precipitation in the regions between 30 ºN and 45 ºN over the Mediterranean during periods of maximum orbitally forced-seasonality (Kutzbach et al., 2014). A role of obliquilty forcing along with precession forcing, in increasing the seasonality and influencing Mediterranean winter rainfall has also been proposed (Bosmans et al., 2015a).

There is ample evidence suggesting that North African precipitation was at a maximum during the mid-Holocene and during other interglacials (Ziegler et al., 2010; Rohling et al, 2015 for a complete review). Maximum Northern Hemisphere seasonality (summer perihelion–increased insolation; winter aphelion–decreased insolation) has been linked to intensified summer monsoon rainfall over North Africa and also increased Mediterranean storm tracks precipitation in winters (Kutzbach et al., 2014). The analysis of PMIP3 simulations carried out in this study also demonstrate intensified African summer monsoon rainfall through the mid-Holocene, during times of enhanced winter precipitation (Fig. S8). This is consistent with recent proxy reconstructions from northcentral Mediterranean where wet winters tend to occur with high contrasts in local, seasonal insolation and in phase with a vigorous African summer monsoon (Wagner et al., 2019)."

Specific points: Abstract: P. 1

line 22: North Atlantic climatic processes is rather vague, do the authors refer to atmospheric patterns? or to oceanic circulation?

We have changed 'North Atlantic climate processes' to North Atlantic atmospheric circulation.

P1 line 23: (but also elsewhere, see above and below) Summer monsoon rainfall does not reach directly the Mediterranean Basin.

As this sentence stands now, it seems that monsoon directly contribute to Mediterranean

precipitation. I agree that monsoon rain contribute to Mediterranean Sea water through Nile (and fossil river system from N Africa) discharge, but it has to be clearly explained.

Reply: We have now reworded the sentence to read "The hydrological budget of the Mediterranean basin is controlled primarily by two phenomena – the latitudinal migration of the Inter-tropical Convergence Zone and the North Atlantic atmospheric circulation. While the former controls African summer monsoon rainfall that drains into the Mediterranean basin via North African rivers, the latter drives the wintertime storm tracks into the western Mediterranean."

1-Introduction:

P2 line 16 Hydrological not hydrologic

This has been corrected now.

p3 line 1 There is a typo in interglacials

This has been corrected now.

p3-line 3-9 this part reads odd. Please rephrase. I guess the words between "Mediterranean" and" for" should be moved after "(Railsback et al., 2015)", also it is not clear which papers refer to Holocene and which to the LIG (e.g. Zanchetta et al., 2007 is Holocene, not LIG).

This has been rephrased now and reads "Proxy and model studies suggest increased winter rainfall in the Mediterranean during the Holocene (Carrión, 2002; Fletcher and Sánchez Goñi, 2008; Magny et al., 2011, 2013; Peyron et al., 2011; Zanchetta et al., 2007; Zielhofer et al., 2017) and also during the MIS (Marine Isotope Stage) 5e (Drysdale et al., 2005; Milner et al., 2012; Regattieri et al., 2014) i.e. the warm periods of the last interglacial (Railsback et al., 2015).

p3 line 20 to the end of the section: it should be moved in a paragraph of site description or in material and methods, it is not introduction.

We have now moved the sentence to section 2 Materials and Methods.

IMPORTANT: a sentence clearly explaining the aim of the paper is missing from the introduction, please add it at the end.

We have now modified the last part of the Introduction to clearly state the objective of this study. It now reads:

"Mediterranean climates are characterized by strong seasonal contrasts with dry summers and wet and highest precipitation amounts annually during winters from October to March. This winter rain is highly variable in amplitude (Xoplaki et al., 2004). Changes in winter rainfall are critical for regional socioeconomic development for the Mediterranean region, but there still remains a lot of ambiguity on the pattern and mechanism of winter rainfall variability, specifically on Quaternary timescales (IPCC, 2014). Proxy and model studies suggest increased winter rainfall in the western Mediterranean and northern Mediterranean borderland during the Holocene (Carrión, 2002; Fletcher and Sánchez Goñi, 2008; Magny et al., 2011, 2013; Peyron et al., 2011; Zanchetta et al., 2007; Zielhofer et al., 2017) and also during the MIS (Marine Isotope Stage) 5e (Drysdale et al., 2005; Milner et al., 2012; Regattieri et al., 2014) i.e. the warm period of the last interglacial (Railsback et al., 2015). A putative link between high seasonality and increased winter rainfall in the central Mediterranean has also been suggested for the MIS 5e (Milner et al., 2012). This increased winter precipitation in the Mediterranean is attributed to higher air–sea temperature difference and locally induced convective precipitation that dominate changes in the freshwater budget on obliquity timescales (Bosmans et al., 2015; Rohling et al., 2015; references therein). Alternatively, recent study from the oldest lake in Europe, Lake Ohrid in the Balkan peninsula show that high North Atlantic sourced moisture into the Mediterranean during winters was the primary driver of Mediterranean hydrological changes both on precessional and seasonal timescales during the interglacials of the past 1.36 Ma (Wagner et al., 2019). The Atlantic signature in Mediterranean precipitation is also visible during MIS 10-11 in a leaf-wax isotopic record from Tenaghi Philippon peatland, NE Greece (Ardenghi et al., 2019). In this light, direct rainfall/sea surface salinity (SSS) and sea surface temperature (SST) estimates from well-located, regionally representative archive that cover multiple interglacial periods, is key to addressing long-standing questions regarding the underlying mechanisms and amplitude of winter precipitation variability.

In this study, we bridge the gap by investigating SSS and SST changes in the marine sediment core GDEC-4-2 located off eastern Corsica (Fig. 1). Previously, Toucanne et al. (2015) used sediment characteristics from GDEC-4-2 to propose enhanced North Atlantic-sourced rainfall in the Western Mediterranean during warm intervals of interglacial periods over the last 547 ka BP. Here, we develop independent geochemical record to assess

precipitation variability by reconstructing runoff (rainfall/salinity) and temperature changes at the GDEC-4-2 site during the Holocene (MIS 1), the last (MIS 5) and penultimate (MIS 7) interglacials using trace element and stable isotopes of the planktonic foraminifera Globigerina bulloides (Fig. 1). We then compare our geochemical proxy records with PMIP model simulations and prominent Mediterranean records, to provide a mechanistic understanding of interglacial precipitation variability."

2 Material and methods

p4 line 14 GDEC is WAS RECOVERED from

This has been corrected now.

P4 line 18-20 Rephrase, a sediment core cannot capture variation in storm track (sediment properties yes, but it should be better explained).

We have changed it to read "the composition of the sediment core acquired"

2.1 Stable isotope analyses Which was the previous resolution of stable isotope analyses? which is the new one? There are not enough details about analytical method (i.e. which calibration method has been used?, which is the reaction time? If analytical methods were the same as in the Toucanne et al paper, it should be stated clearly.

We have now added specific details about what was published in Toucanne et al., and new in this manuscript. Line 7-11 Page 5 now read "We use previously published stable isotopic results from Toucanne et al., 2015 for the Holocene and MIS 7c and 7e period. For MIS 5e period, we sampled G. bulloides for isotopic analysis as G.bulloides isotopic data for this interval is not used in Toucanne et al., 2015. The temporal resolution for stable isotope data ranges from ~0.2 - 3ka BP/per measurement."

2.2 Trace elements analyses Add the resolution (spatial) at which these analyses were done.

Reply: The temporal resolution for stable isotope data ranges from ~0.2 - 3ka BP/per $\delta_{18}O$ measurement and for the trace element ranges from ~0.2- 2 ka BP/per analysis. Line 19-21, Page 5.

p.5 line 12 proxy data OBTAINED FROM IT are representative

This has been corrected now.

p5 line 13 30 m, and are reflect THUS REFLECTING surface: : :

This has been corrected now.

p5 line 25 TO CHECK FOR INTERNAL CONSISTENCY recurrent analyses: : :

This has been corrected now.

p5 line 26 precession PRECISION p6 line 13 "comparable" to the TO THAT OBSERVED IN previous studies

This has been corrected now.

p6 line 15 Here it is stated that others core top in the Mediterranean have lower Ba/Ca values so it is not clear if Ba/Ca values observed here are comparable or not with previous studies: : :. Also the sentence about calibration for used to infer temperature from Mg/Ca (line19-21) should be better separated by the discussion about Ba/Ca and better motivated. It is the same calibration used in previous studies in the region or not? (i.e. from where the McConnell calibration comes from?)

We have now added more details about the Ba/Ca data and the calibration used which now reads

" The range of Ba/Ca in G. bulloides observed in this study is higher than other planktonic species but is comparable to that observed in previous studies on G. bulloides from Mediterranean and other regions, such as, Ba/Ca values in G. bulloides calcite is reported to be significantly higher than other planktonic species collected from core tops across the Mediterranean (Ferguson et al., 2008; Sprovieri et al., 2008). Marr et al., (2013) reported Ba/Ca to range between ~8-14 $\mu$mol/mol in G. bulloides collected from core tops in southwestern Pacific in the Tasman Sea off New Zealand. Previously, Lea and Boyle, (1991) suggested that several planktonic foraminifera species have high Ba/Ca ratios owing to the differences in the way these foraminifera precipitate their shells."

p6 line 22-23 for this region of the Mediterranean P6 line 24 generally

This has been corrected now.

p6 line 26 records TO for our core site that constrain river runoff AT OUR SITE

This has been corrected now.

2.3 PMIP3 model simulation

See general comments, it is a non-sense to use a mid-Holocene simulations to infer
mechanisms working for other interglacials characterised by different boundary conditions.
I would almost remove this part: : :please instead consider modelling results from
Bosmans et al. 2015 paper.

We would like to disagree with the reviewer on this point. The model simulations used in our
study to infer mechanisms for winter rainfall is for the mid-Holocene, which is in fact one of
the interglacials for which we have used Ba/Ca as runoff proxy. The boundary conditions for
the last and penultimate interglacial were indeed different and we need PMIP4 ensemble to
understand the mechanisms for the last interglacial. Nonetheless, recent proxy and transient
climate model simulations suggest similar mechanism plying during the past interglacials of
1.3 Ma, further attesting our hypothesis.  Please refer to earlier replies to comment 2, for
detailed explanation of why we have retained the model simulation section as a part of the
manuscript. However, as the reviewer suggested we have have largely modified the entire
section to include previous published literature and also some  recent publications on the
Mediterranean winter rainfall  (for example, Wagner et al., 2019 Nature; Ardhengi et al.,2019
QSR; Marzocchi et al., 2019 Paleoceanography and Paleoclimatolgy).

2.4 Chronology As the GDEC record has been published already and now the chronology is
updated by aligning to the Marino et al. (2015) curve
I suggest the authors to quantify the difference with the previously published record.
Last, at the end of this paragraph authors should insert the resulting temporal resolution for
both the stable isotope and the trace element records.
Reply: With the new chronology for the last interglacial, the difference in the warm interval
MIS 5e observed in new chronology and Toucanne et al., chronology, is ~3ka BP.
The temporal resolution for stable isotope data ranges from ~0.2 - 3ka BP/per $\delta_{18}O$
measurement and for the trace element ranges from ~0.2- 2 ka BP/per analysis.
This section now reads " We use previously published stable isotopic results from Toucanne
et al., 2015 for the Holocene and MIS 7c and 7e period. For MIS 5e period, we sampled G.
bulloides for isotopic analysis as G.bulloides isotopic data for this interval is not used in
Toucanne et al., 2015. The temporal resolution for stable isotope data ranges from ~0.2 - 3ka

BP/per measurement. Oxygen and carbon isotope ratios were measured using Thermo Scientific Delta V plus Isotope Ratio Mass Spectrometer fitted with a GasBench II preparation and introduction device, operated by Pôle Spectrométrie Océan (PSO, IFREMER, IUEM, CNRS), located at the Institut Universitaire Européen de la Mer (IUEM / UBO) at Plouzané, France (For more details on the analytical methods, please refer to Toucanne et al., 2015)."

p7 line 25: to exploit EXPLOITING
This has been corrected now.

3 Results
3.1 Proxy systematics
 p.8 line 10 "_18O OF in foraminifera" and "and BY
_18Osw". Also, you should put a reference here and also quote Fig. S3.
We have put a reference and referred to Figure S3.

p.8 line 14. Sentence not clear. Also the _18O of the river water is related to P/E ratio, not only the _18O of the sea water. At the end of the page you should quote the relative supplementary text and figure.
We have now reworded the sentence and also referred to the correct supplementary figure.

3.2 SeaWater oxygen isotope and Mg/Ca based SSTs
p.9 line2 highER, not high and also quote a figure after periods
This has been corrected now and figures have been quoted.

p.9 line 4 and THESE INTERVALS ARE also characterized
This has been corrected now.

p.9 line 16 BP, with AND BY lowest values Authors should briefly comment here about MIS7 temperatures and removing the relative paragraph, which is really confused, from the supplementary material (see specific comments to Supp Mat).
We have now given details of MIS7 temperature variability which now reads "These globally warm periods are characterized by increased Mg/Ca-based-SSTs at GDEC-4-2 with ~18 ℃ Holocene values, and MIS 5e being the warmest with temperatures averaging ~24℃,

although at this site, our SST reconstructions indicate increased riverine discharge during MIS 7c and 7e led to local SST being cooler (Fig. S4)."

3.3 Precipitation and salinity changes inferred from foraminifera Ba/Ca

p.9 line 27 Why the increase abundance of benthic foraminifera indicates an increase of OM transportation to the bottom?

Reply: Increased abundance of benthic foraminifera suggest significant input of OM occurred at GDEC 4-2 site during periods of sea-level highstands. This increased exportation of organic matter is related to enhanced Golo river discharge and increased productivity exported to seabed.

4 Discussion

p10 line 11 Last Interglacial (here and after)

This has been corrected now.

p10 line 15 which delivered DELIVERING

This has been corrected now.

p.10 line 16 As above, you should specify that the monsoonal rain is delivered by the Nile and by -now fossil- river system in the North Africa

We have now specified the contribution of fresh water from Nile and other North African rivers into the Mediterranean which now reads "In the eastern Mediterranean Sea at site LC21 located in the Aegean Sea, low $\delta_{18}$O foraminifera values during the last interglacial following the TII (Fig. 3g) has been attributed to intensified North African Monsoon as Northern Hemisphere insolation peaked and the ITCZ moved northward, delivering large amounts of freshwater via Nile river and other North African rivers into the eastern Mediterranean around ~128–122 ka BP (Rodríguez-Sanz et al., 2017)."

p10 line 20 waters FROM WHICH the foraminifera calcite FORMS

This has been corrected now.

p10 line 25 increased LOCAL precipitation

This has been corrected now.

p11 line 4 put a comma after Ba/Ca and another one after MIS5e p11 line 5 synchronous TO WETTER CONDITIONS INFERRED

This has been corrected now.

p11 line 7 as above, Zanchetta et al., 2007 is Holocene and not LIG, I guess Regattieri et al., is 2014 or 2017 and not 2015.

We have corrected the references now.

Lines 9-10 are a repetition of lines 5-6.

We have now merged these two lines.

p11 line 14 What does "regional sedimentary signal means"???

We mean the regional precipitation pattern in the western Mediterranean. We have changed it now, which now reads "Recently, Pasquier et al. (2019) reported episodes of enhanced proportion of land-derived material suggesting significant increase in precipitation amount over the Gulf of Lion catchment area during the warm intervals of both Holocene and MIS5, further attesting the regional precipitation pattern in the western Mediterranean and its northern borderlands."

p11 line 17 Tzedakis et al, 2007 does not report any Holocene pollen record showing higher seasonality and for should be FROM sites

We have now corrected the reference to Lawson et al., 2005 for the Holocene.

In general, what is new in this paragraph with respect to the Toucanne et al paper???

Our geochemical records for past SSS and SST support the hypothesis proposed using by Toucanne et al., 2015 using indirect sedimentological proxies for rainfall, and add greater confidence to their findings. We have clearly pointed this out in the introduction and also in this paragraph, which now reads:

"Together our site GDEC4-2 and other discussed western Mediterranean sites lie outside the influence of the ITCZ-controlled African summer rainfall suggesting that these archives record enhanced winter rainfall during the Holocene and the last interglacial. Interestingly,

our geochemical records also show that increased wintertime precipitation and lower SSS in the western Mediterranean extended as far back as the warm intervals of penultimate interglacial, MIS-7c and 7e, corroborating with previous sedimentological work by Toucanne et al., 2015 (Fig. 4e). These results therefore support the hypothesis that high rainfall during interglacials was a distinctive feature of Mediterranean climate (Sierro et al., 2000; Valero et al., 2014; Bosmans et al., 2015a, Wagner et al., 2019), confirming by extension that the precession minima (boreal summer insolation maxima and winter minima) paced rainfall variability."

4.3 Contribution of western Mediterranean precipitation in sapropel deposition (in should be TO instead of in) Toucanne et al paper's speaks about an increase of western Mediterranean storm track, not about an increase in North Atlantic sourced precipitation during period of sapropel deposition. I agree that wMed precipitation play a role in triggering anoxia and sapropel deposition and I do not support as well the Rohling hypothesis. However, this part is very confused and I do not see any reason to invoke an increase of moisture transport from the North Atlantic. This claim is not supported by the references provided in lines 19-20, nor by the new presented data, and is in contrast with what already proposed basing on GDEC data.
Reply: We have discussed this in our previous reply.

p14 line 1 there's a typo in supported (or proposed?)
We would like to point that it is the word 'purported' and not a typo.

p14 line 12 how mid-latitude storm tracks can contribute to organic fluxes? this sentence has no sense.
Reply: Mid-latitude storm tracks bring increased rainfall that contributes to freshwater via rivers and bring organic matter into the Mediterranean. At GDEC -4-2, this increased exportation of organic matter flux is therefore related to enhanced Golo river discharge and increased productivity exported to seabed. This now reads "Such mid-latitude storm tracks originating from the North Atlantic contributed to increased freshwater via runoff and organic fluxes into the Mediterranean Sea. This in turn maintained the already-disrupted hydrology of the Mediterranean, and reduced the intermediate and deep-water ventilation."

Conclusion: they need to be largely rewritten following provided comments:

We have modified the conclusions now. It now reads

"In this study, we used geochemical proxies to better assess the variation in winter rainfall in the western Mediterranean during the Holocene and the past two interglacials. Our geochemical data suggest increased runoff/ rainfall during the warm periods of the Holocene and the past two interglacials. Proxy data demonstrate that the intensity of the precession-controlled wintertime rainfall in the western Mediterranean was modulated by eccentricity, with times of high eccentricity characterised by higher rainfall and river outflow. These results along with the analysis of Holocene climate simulations support increased winter precipitation sourced from the North-Atlantic in a warmer western Mediterranean during the past. Our data and model results also show that high rainfall events in the northern Mediterranean borderland occurred at times of intensified North African summer monsoon and the sapropel deposition in the Mediterranean basin. This is in agreement with recently published proxy reconstructions for past 1.3 Ma and climate simulations from Lake Ohrid in Central Mediterranean (Wagner et al., 2019). The close chronological correspondence of increased river outflow and winter rainfall to organic carbon deposition and sapropel occurrence supports a causal link. We suggest a close coupling between low and high latitude atmospheric-oceanic processes in triggering anoxia in the basin, with a contribution from, both Nile River outflow changes due to variations in African summer monsoon rainfall as well as North Atlantic climatically-controlled winter-rainfall driving outflow changes in the western Mediterranean."

Figures They are all rather poorly constructed in my opinion and need to be largely modified. I suggest to prepare a proper results figure showing only the results from GDEC for all the period discussed (this should be fig. 2 not 3), then to make others figures with the three intervals separated and where the records used for comparison have to be shown. Please enlarge all the figure and be sure that axes's values are appropriated. Figure 4 is useless in my opinion, all the mentioned sites needs to be shown in fig. 1

We have now enlarged all the figures and also introduced a new Figure 2, which has the results for GDEC-4-2 only, as suggested by the reviewer. Subsequently, old figure 2 is now Figure 3, which shows the comparison of Last interglacial studies and GDEC-4-2 results on a new age scale from Marino et al., (2015). Original Figure 3 is now Figure 4 showing comparison of all the three interglacials with other studies. We have also modified Figure 4 and added recent pollen data and weight% Total Inorganic Carbon data from Lake Ohrid in

the Central Mediterranean for the studied intervals, which is in line with our geochemical data and also supports our model output (Wagner et al., 2019, Nature). We have kept the model simulation results (original Figure 4 now Figure 5) as part of our article, as we think this is important to understand the source of winter rains and complements other model and isotopic studies tracking the source of moisture from the North Atlantic into the Mediterranean, as also discussed above in detail. Additionally, we have also modified Figure 1 and its caption and added lake level records used in Figure 4. New Figures 1 and 2 are shown at the end of this document.

Fig. 1 The line indicating the Mediterranean storm tracks has no sense, this line may resemble the major trajectory of North Atlantic storm track, but it seems to me an over simplification.
The arrow indicating Mediterranean storm tracks show the general direction and path of the storm track and is not by scale.

Argentarola cave is not mentioned in the text, why it is mentioned here? From where the position of the ITCZ comes from? again it seems poor and over simplified.
We agree with the reviewer and Argentarola cave has been taken out from Figure 1 now. 'The position of ITCZ is the maximum northward displacement over the last million years (Tuenter et al., 2003)'. We have now added this information in the caption of Figure 1.

Please put all the reference for the terrestrial and marine sites in the caption of Figure 1, this would avoid the whole first paragraph of supplementary text, which is really confused and not useful at all.
We have now moved all the references for terrestrial and marine sites in Figure 1 caption. We have also added the location of sites in the Figure.

Fig.2 It should report only the results from GDEC, whereas all the other records used for comparison should be moved to another figure (fig.3)
We have now changed Fig. 2 to only show our stable isotopes and trace element data obtained in this study. Fig. 3 is now comparison of our results with previously published studies.

Supplementary information The first two paragraph (regarding the records used for

comparison and the one regarding the MIS7 temperature, should be shorten and accommodated in the main text and in figure captions as indicated in previous comments.

We have now included the first paragraph detailing the sites information in Figure 1 and have moved the relevant text about MIS7 in the main manuscript.

Fig. S5: Why there are only 3 points if in table s5 five sampling points are reported? The high correlation coefficient reported is simply an artefact due to the very limited number of points!

Indeed, the high correlation is a result of limited data points, which is why we have refrained from making large claims on the reconstructed salinity and this discussion is in the supplementary information.

**Modified Figure 1 and new figure 2:**

[Figure]

**Figure 1: Location of GDEC-4-2 (red) in the northern Tyrrhenian Sea and other marine (blue) and terrestrial archives (green). Numbers and black dots denote the lake level records used to compare results with model simulations in Figure 5. 1) Lake Medina in southern Spain (Reed et al., 2001); (2) Lake Siles in southern Spain (Carrión, 2002); (3) Lake Cerin (Magny et al., 2011); (4) Lake Ledro in northern Italy (Magny et al., 2012); (5) Lake Accesa in central Italy (Magny et al., 2007); (6) Lake Grande diMonticchio in Basilicata, southern Italy (Allen et al., 1999); (7) Lake Albano and Lake Nema (Ariztegui et al., 2000); (8) Lake Preola in Sicily (Magny et al., 2011); (9) Lake Xinias in northern Greece (Digerfeldt et al., 2007); (10) Lake Golhisar in south-western Turkey (Eastwood et al., 2007); (11) Lake Eski Acigol in central Turkey (Turner et al., 2008); (12) Lake Van in Turkey (Pickarski and Litt, 2017). Red band and red dotted line denotes the extent of modern summer ITCZ and the maximum northward reach of ITCZ in the past respectively. Also shown are the sea-level pressures in North Atlantic and the direction of Mediterranean storm tracks (black).**

[Figure]

**Figure 2: GDEC-4-2 results for the last three interglacials. (a) δ18O *G. bulloides*; (b) Mg/Ca-based SSTs from *G. bulloides* (blue); (c) Ba/Ca in foraminifera as a proxy of river discharge. Vertical light yellow bars indicate interglacial conditions *s.l.* and dark yellow bars denote interglacials warm intervals and the interglacial *s.s.* Sapropel deposition intervals, Heinrich stadials (blue bar) and mid-points of glacial terminations (dashed red line, following Barker et al., 2019) shown on top.**